# The impact of continuous quality improvement on coverage of antenatal HIV care tests in rural South Africa: Results of a stepped-wedge cluster-randomised controlled implementation trial

H. Manisha Yapa[1,2]*, Jan-Walter De Neve[3], Terusha Chetty[4], Carina Herbst[2], Frank A. Post[5], Awachana Jiamsakul[1], Pascal Geldsetzer[3,6], Guy Harling[2,7], Wendy Dhlomo-Mphatswe[8], Mosa Moshabela[2,9], Philippa Matthews[2,10], Osondu Ogbuoji[11], Frank Tanser[2,9,12,13], Dickman Gareta[2], Kobus Herbst[2], Deenan Pillay[2,14], Sally Wyke[2,15], Till Bärnighausen[2,3,7,16,17]

1 The Kirby Institute, University of New South Wales Sydney, NSW, Australia, 2 Africa Health Research Institute (AHRI), KwaZulu-Natal, South Africa, 3 Heidelberg Institute of Global Health (HIGH), Medical Faculty and University Hospital, Heidelberg University, Heidelberg, Germany, 4 Health systems Research Unit, South African Medical Research Council, Durban, South Africa, 5 King's College Hospital NHS Foundation Trust, London, United Kingdom, 6 Division of Primary Care and Population Health, Department of Medicine, Stanford University, Stanford, California, United States of America, 7 Institute for Global Health, University College London, London, United Kingdom, 8 School of Clinical Medicine, Discipline of Obstetrics and Gynaecology, University of KwaZulu-Natal, Durban, South Africa, 9 School of Nursing and Public Health, University of KwaZulu-Natal, Durban, South Africa, 10 Islington GP Federation, London, United Kingdom, 11 Global Health Institute, Duke University, Durham, North Carolina, United States of America, 12 Lincoln International Institute for Rural Health, University of Lincoln, Lincoln, United Kingdom, 13 Centre for the AIDS Programme of Research in South Africa (CAPRISA), University of KwaZulu-Natal, Durban, South Africa, 14 Division of Infection and Immunity, University College London, London, United Kingdom, 15 Institute for Health & Wellbeing, University of Glasgow, Glasgow, United Kingdom, 16 MRC/Wits Rural Public Health and Health Transitions Research Unit (Agincourt), Faculty of Health Sciences, University of the Witwatersrand, Johannesburg, South Africa, 17 Department of Global Health and Population, Harvard T.H. Chan School of Public Health, Boston, Massachusetts, United States of America

* myapa@kirby.unsw.edu.au

**Data Availability Statement:** Access to the study data is governed by the data access policy at the

## Abstract

### Background

Evidence for the effectiveness of continuous quality improvement (CQI) in resource-poor settings is very limited. We aimed to establish the effects of CQI on quality of antenatal HIV care in primary care clinics in rural South Africa.

### Methods and findings

We conducted a stepped-wedge cluster-randomised controlled trial (RCT) comparing CQI to usual standard of antenatal care (ANC) in 7 nurse-led, public-sector primary care clinics—combined into 6 clusters—over 8 steps and 19 months. Clusters randomly switched from comparator to intervention on pre-specified dates until all had rolled over to the CQI intervention. Investigators and clusters were blinded to randomisation until 2 weeks prior to each step. The intervention was delivered by trained CQI mentors and included standard CQI tools

AHRI, the institution where this study took place. Access will be granted upon request. Such requests can be made to the AHRI Data Repository via the AHRI website https://www.ahri.org/research/. Researchers who meet the criteria for access to fully anonymised patient level data will be granted data access.

**Funding:** The MONARCH project was co-funded by the Delegation of the European Commission to South Africa, EuropeAid/134286/L/ACT/ZA, and by the Wellcome Trust (through core funding to AHRI). The contents of this document are the sole responsibility of the authors and their affiliated institutions, and can under no circumstances be regarded as reflecting the position of the European Union. AHRI receives core funding from the UK Wellcome Trust grant 082384/Z/07/Z and Howard Hughes Medical Institute. The AHRI Population Intervention Platform is partially funded by the South African Population Research Infrastructure Network (SAPRIN), South African Department of Science and Technology. HMY is supported by an Australian Government Research Training Program (RTP) Scholarship, University of New South Wales, Sydney, Australia. The Kirby Institute is funded by the Australian Government Department of Health and Ageing, and is affiliated with the Faculty of Medicine, UNSW Sydney. JWDN is supported by the Alexander von Humboldt Foundation. TC was supported by the South African Medical Research Council Clinician Researcher Programme. PG was supported by the National Center for Advancing Translational Sciences of the National Institutes of Health (NIH) under Award Number KL2TR003143. TB is supported by the Alexander von Humboldt Professor award, funded by the Federal Ministry of Education and Research; the Wellcome Trust; the Eunice Kennedy Shriver National Institute of Child Health and Human Development (NICHD) of NIH (R01-HD084233), National Institute on Ageing (NIA) of NIH (P01-AI112339), as well as the Fogarty International Center (FIC) of NIH (D43-TW009775). The funders had no role in study design, data collection and analysis, decision to publish, or preparation of the manuscript.

**Competing interests:** The authors have declared that no competing interests exist.

**Abbreviations:** AHRI, Africa Health Research Institute; ANC, antenatal care; ART, antiretroviral therapy; CONSORT, Consolidated Standards of Reporting Trials; CQI, continuous quality improvement; CRH, Centre for Rural Health; DoH, South African National Department of Health; DSMB, Data Safety and Monitoring Board; eMTCT, elimination of mother-to-child transmission of HIV; ICC, intracluster correlation coefficient; IHI,

(process maps, fishbone diagrams, run charts, Plan-Do-Study-Act [PDSA] cycles, and action learning sessions). CQI mentors worked with health workers, including nurses and HIV lay counsellors. The mentors used the standard CQI tools flexibly, tailored to local clinic needs. Health workers were the direct recipients of the intervention, whereas the ultimate beneficiaries were pregnant women attending ANC. Our 2 registered primary endpoints were viral load (VL) monitoring (which is critical for elimination of mother-to-child transmission of HIV [eMTCT] and the health of pregnant women living with HIV) and repeat HIV testing (which is necessary to identify and treat women who seroconvert during pregnancy). All pregnant women who attended their first antenatal visit at one of the 7 study clinics and were $\geq$18 years old at delivery were eligible for endpoint assessment. We performed intention-to-treat (ITT) analyses using modified Poisson generalised linear mixed effects models. We estimated effect sizes with time-step fixed effects and clinic random effects (Model 1). In separate models, we added a nested random clinic–time step interaction term (Model 2) or individual random effects (Model 3). Between 15 July 2015 and 30 January 2017, 2,160 participants with 13,212 ANC visits (intervention $n = 6,877$, control $n = 6,335$) were eligible for ITT analysis. No adverse events were reported. Median age at first booking was 25 years (interquartile range [IQR] 21 to 30), and median parity was 1 (IQR 0 to 2). HIV prevalence was 47% (95% CI 42% to 53%). In Model 1, CQI significantly increased VL monitoring (relative risk [RR] 1.38, 95% CI 1.21 to 1.57, $p < 0.001$) but did not improve repeat HIV testing (RR 1.00, 95% CI 0.88 to 1.13, $p = 0.958$). These results remained essentially the same in both Model 2 and Model 3. Limitations of our study include that we did not establish impact beyond the duration of the relatively short study period of 19 months, and that transition steps may have been too short to achieve the full potential impact of the CQI intervention.

## Conclusions

We found that CQI can be effective at increasing quality of primary care in rural Africa. Policy makers should consider CQI as a routine intervention to boost quality of primary care in rural African communities. Implementation research should accompany future CQI use to elucidate mechanisms of action and to identify factors supporting long-term success.

## Trial registration

This trial is registered at ClinicalTrials.gov under registration number NCT02626351.

## Author summary

### Why was this study done?

- Gaps in implementation of evidence-based guidelines can slow progress towards major health systems goals, such as the elimination of mother-to-child transmission of HIV (eMTCT).

- Continuous quality improvement (CQI) has the potential to improve service delivery with available resources and has been successful in resource-rich settings.

- There is very limited evidence on the impact of CQI in improving health systems in resource-poor primary care.

Institute for Healthcare Improvement; IQR, interquartile range; ITT, intention-to-treat; MONARCH, Management and Optimisation of Nutrition, Antenatal, Reproductive, Child health & HIV care; MTCT, mother-to-child transmission of HIV; PDSA, Plan-Do-Study-Act; PIPSA, AHRI Population Intervention Platform Surveillance Area; RCT, randomised controlled trial; REDCap, Research Electronic Data Capture; RR, relative risk; TIDieR, Template for Intervention Description and Replication; VL, HIV viral load.

## What did the researchers do and find?

- We assigned 7 public-sector primary care clinics in rural South Africa to receive CQI in random sequence.

- The intervention was delivered by a trained team of local CQI mentors who worked collaboratively with clinic health workers. The mentors and health workers aimed to address root causes of failure to perform HIV viral load (VL) monitoring among pregnant women living with HIV and repeat HIV testing among pregnant women not living with HIV.

- CQI increased HIV VL monitoring by nearly 40% but did not improve repeat HIV testing.

- Lay counsellors, who conducted HIV testing and counselling in the primary care clinics, were redeployed during the study period, placing a strain on available human resources for HIV testing.

## What do these findings mean?

- CQI can be effective at improving quality of primary care in resource-poor communities.

- One explanation for the different effectiveness of CQI on our 2 primary endpoints is that health workers may have perceived VL monitoring to be more important for health outcomes than repeat HIV testing.

- Resource shortages, particularly staffing shortages, may have prevented CQI from achieving its full potential.

- Future research should focus on the conditions under which CQI is most likely to be successful.

## Introduction

Continuous quality improvement (CQI) is an important approach to improving quality of care and adherence to clinical guidelines in the health sector [1,2]. Originally developed to streamline production processes in the consumer industry [3], CQI was adopted in the 1990s by the healthcare sector to improve organisational systems to create better quality of care and health outcomes [4]. CQI focuses on developing healthcare providers' capacity to improve quality of care. It consists of a set of adaptable but systematic techniques to diagnose quality problems using real-time data [1,2] and to address these problems with existing resources [5]. These properties make CQI an attractive approach to improve primary care in resource-poor countries and communities, such as in sub-Saharan Africa [6]. Indeed, CQI is being increasingly rolled out at national scale in several countries in sub-Saharan Africa, with the aim to improve quality of care in primary care [7–11]. The focus on quality improvement in primary care is important, because a primary care visit is often the first and only contact people have with their local health system when seeking care [12].

   To date, there have only been 2 randomised controlled trials (RCTs) that have tested whether CQI improves quality of care in the primary care systems in sub-Saharan Africa, one

in Malawi [13] and the other one in Nigeria [14]. Both trials found CQI to be ineffective [13,14]. One possible reason for the lack of effectiveness in these 2 trials is that the investigators evaluated CQI in terms of health or distal healthcare utilisation outcomes—neonatal and perinatal mortality rates in the study in Malawi [13] and 6-month postpartum retention in the study in Nigeria [14]. These endpoints are important, and they are rigorously assessed in the 2 studies. However, these endpoints are quite distal from the CQI activities that took place in the primary care clinics. The variability of these endpoints is therefore likely largely a function of factors outside the sphere of influence of CQI, such as emergency care availability, road infrastructure, and mothers' educational attainment.

For our RCT of CQI effectiveness, we thus decided to use 2 primary endpoints that are closely and directly linked to CQI activities. Our trial took place in an HIV hyperendemic community in rural South Africa, where antenatal care (ANC) in primary care fulfils 2 important HIV care–related functions: to ensure that all pregnant women living with HIV are (1) identified and (2) successfully treated with antiretroviral therapy (ART). We choose measures that capture the procedural quality of care of these 2 key ANC functions as primary endpoints: viral load (VL) monitoring and repeat HIV testing. Both endpoints are important measures of ANC quality and are necessary conditions for preventing mother-to-child transmission of HIV (MTCT) [15,16]. Ultimately, the purpose of healthcare is to improve health outcomes. Implementation science, however, focuses on the improvement of processes whose intermediate outcomes, such as testing coverage, can be directly influenced by changes to health services and are known to be strong determinants of health outcomes [17].

To capture proximate effects of CQI relevant to successful ART, we chose VL monitoring, because VL is the marker of ART response—and MTCT risk—prescribed in the South African national guidelines for ART and for the elimination of MTCT (eMTCT) [18, 19]. Regular VL monitoring enables timely management of virologic failure to achieve eMTCT, including treatment switches and enhanced antiretroviral prophylaxis for HIV-exposed infants [20–22]. Despite the importance of VL monitoring and clear stipulation in the national guidelines [23], most pregnant women on ART are not regularly monitored for VL [24,25].

To capture proximate effects of CQI relevant to the detection of HIV infection, we chose repeat HIV testing, because incident HIV infection in pregnancy increases risk of MTCT as VL peaks shortly after infection [26]. In South Africa, where antenatal HIV incidence is high [27,28], early diagnosis of incident HIV through repeat testing during pregnancy is a critical step towards eMTCT [29]. Yet national guideline recommendations for retesting pregnant women who tested HIV negative during a prior HIV test in ANC are not well adhered to [26].

Our stepped-wedge cluster RCT aimed to establish the effectiveness of CQI on important proximate indicators of the quality of HIV-related ANC in primary care. We hypothesised that a CQI intervention would be effective at improving (i) VL monitoring in pregnant women living with HIV and (ii) repeat HIV testing in pregnant women not living with HIV.

## Methods

Details of the Management and Optimisation of Nutrition, Antenatal, Reproductive, Child health & HIV care (MONARCH) implementation project and study have been previously published [30] and are summarised below.

### Study setting

The Africa Health Research Institute (AHRI) at Somkhele (previously known as the Africa Centre for Population Health) is located in a rural community in northern KwaZulu-Natal, South Africa. Our CQI intervention was conducted at 7 nurse-led South African National

Department of Health (DoH) primary care clinics: 6 were located within the geographic bounds of the AHRI Population Intervention Platform Surveillance Area (PIPSA) South [31], and 1 clinic was located in the market town of Mtubatuba, which is often used by PIPSA residents (Fig 1). Management of the primary care clinics is overseen by Hlabisa Hospital, the local district hospital. HIV prevalence amongst women of reproductive age in this area is approximately 37% [32]. Additional contextual information is described in the Supporting Information: laboratory results workflow (S1 Text), clinic size (S1 Table), and staffing (including lay counsellors; S2 Table). The intervention was delivered by an external CQI team of mentors (including 2 isiZulu-speaking nurses) from the Centre for Rural Health (CRH) at the University of KwaZulu-Natal, who travelled to the study community (hereafter referred to as the CRH team). The mentors were closely supported by an improvement advisor (consultant obstetrician), a scientific advisor, and a data manager.

## Trial design

We conducted a stepped-wedge cluster RCT (www.clinicaltrials.gov; NCT02626351) from 15 July 2015 to 30 January 2017. Each clinic formed a cluster except for the 2 smallest clinics, which were merged into one cluster. After a 2-month baseline data collection period, the first cluster rolled over to the intervention on 29 September 2015. Each subsequent cluster rolled over from control to intervention in random order every 2 months (Fig 2).

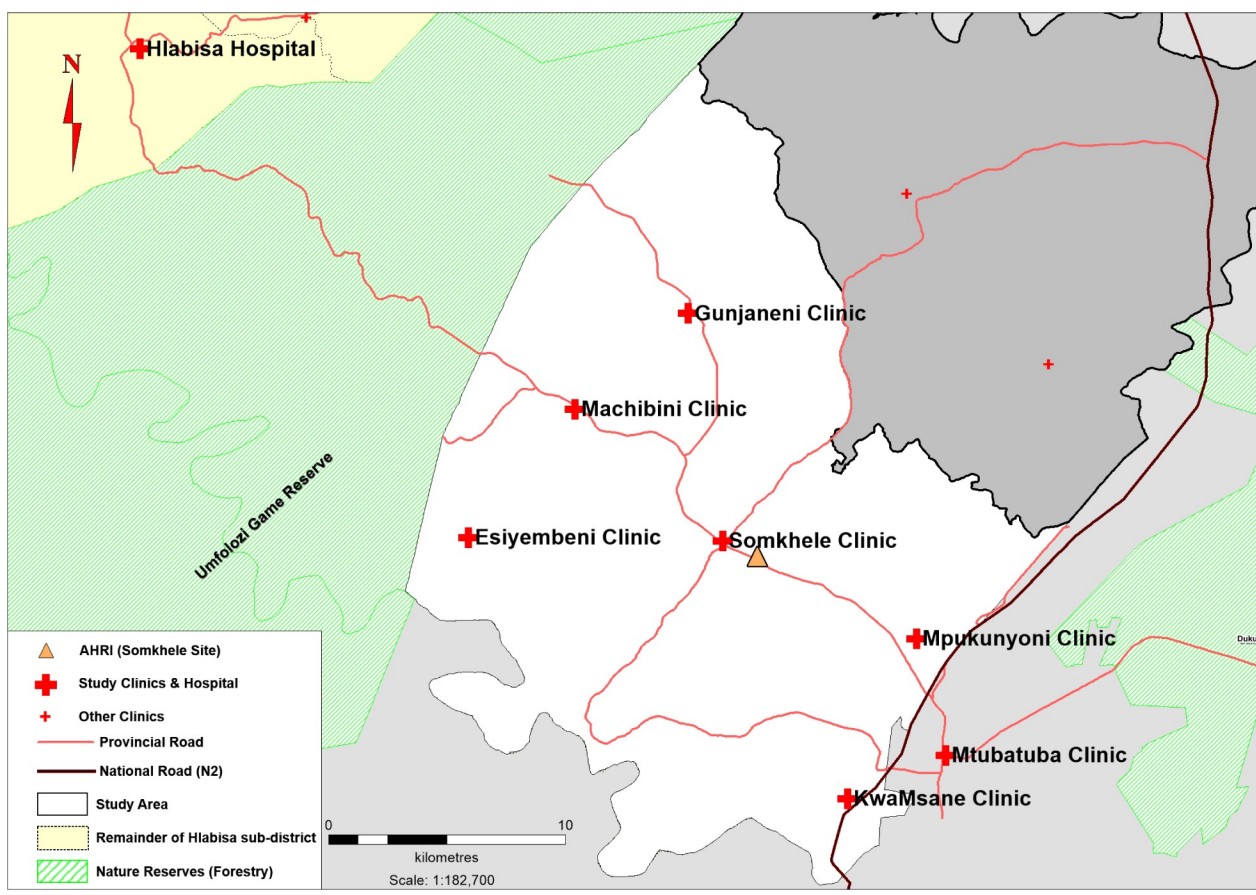

**Fig 1. Participating study clinics located within AHRI PIPSA.** The PIPSA is depicted in white and covers 438 km². Primary care clinics and the local district hospital, Hlabisa Hospital, are marked with a red cross. *Source credit: Sabelo Ntuli, AHRI Research Data Management.* AHRI, Africa Health Research Institute; PIPSA, AHRI Population Intervention Platform Surveillance Area.

| Clicni | Step (Month) | | | | | | | | Total |
|---|---|---|---|---|---|---|---|---|---|
| | 0 (1-8) | 1 (9-10) | 2 (11-12) | 3 (13-14) | 4 (15-16) | 5 (17-18) | 6 (19-20) | 7 (21-25) | |
| 1 | 597 | 197 | 235 | 200 | 266 | 266 | 244 | 230 | 2235 |
| 2 | 1170 | 330 | 419 | 389 | 463 | 424 | 458 | 423 | 4076 |
| 3a | 230 | 90 | 88 | 87 | 74 | 68 | 58 | 42 | 737 |
| 3b | 79 | 22 | 18 | 11 | 13 | 15 | 26 | 44 | 228 |
| 4 | 890 | 372 | 413 | 398 | 461 | 481 | 369 | 349 | 3733 |
| 5 | 265 | 51 | 85 | 69 | 79 | 113 | 101 | 79 | 842 |
| 6 | 336 | 134 | 110 | 138 | 189 | 184 | 147 | 123 | 1361 |
| Total | 3567 | 1196 | 1368 | 1292 | 1545 | 1551 | 1403 | 1290 | 13,212 |

Key: Pre-CQI | CQI implementation | Post-CQI

**Fig 2. Study design and endpoint observations by actual randomisation sequence.** Primary care clinics provided pre-intervention data until each rolled over to the CQI intervention in random order. All clinics provided data continuously throughout the study period. Baseline data collection across all clinics occurred from 15 July 2015 to 28 September 2015 (Step 0). As ANC data were captured retrospectively at delivery, the total observation period exceeded the data collection period by approximately 6 months. Width of each step is proportional to the number of months under observation. The baseline period (pre-intervention, depicted in light blue) contributed approximately 8 months, and the endline (Step 7) contributed approximately 4.5 months [30]. Intervention steps (intensive CQI phase, 2-month step) are depicted in medium blue. ANC, antenatal care; CQI, continuous quality improvement.

Trial registration occurred after the baseline and the first step of this 8-step stepped-wedge RCT, on 10 December 2015. The reason for this timing was that it became clear during the baseline that a rigorous scientific evaluation of the CQI intervention would be feasible and desirable for both government and implementing partners (S1 Text). The stepped-wedge design was selected for both pragmatic and ethical reasons [30]. The description of our results follows the 2018 Consolidated Standards of Reporting Trials (CONSORT) extension for stepped-wedge RCTs [33] (S1 CONSORT Checklist).

## Intervention

We use the Template for Intervention Description and Replication (TIDieR) to describe the intervention in detail (Table 1) [34]. Briefly, the intervention focused on developing the capacity of local ANC health workers in study clinics and aimed to improve implementation of the national eMTCT guidelines. The intervention was based on the Institute for Healthcare Improvement (IHI) breakthrough collaborative CQI model [35]. Clinical processes and resources were first ascertained during situational analyses conducted in the 2-week lead-up to intervention rollover. CQI tools provided a structured approach to improving process change

**Table 1. MONARCH CQI intervention description: TIDieR framework.**

| Name (1) | MONARCH implementation project |
|---|---|
| Why (2) | eMTCT requires rigorous implementation of national treatment and care guidelines while respecting and protecting the human rights of women living with HIV. Yet in resource-poor settings, gaps in guidelines implementation may hinder progress towards eMTCT. CQI has the potential to improve clinical processes at any level of available resources for routine care. It has been shown to support guidelines implementation in resource-rich settings, but rigorous evidence in resource-poor settings is sparse. The CQI intervention was designed to improve implementation of South African eMTCT guidelines in rural primary care clinics, with a focus on improved diagnosis of incident HIV infection (through repeat HIV testing in pregnancy) and better monitoring of HIV VL in pregnancy. |

*(Continued)*

**Table 1.** (Continued)

| What, materials (3) | As a complex behavioural intervention, CQI draws on several impact theories, particularly TQM, NPT, and educational and motivational theories [40]. CQI aims to improve quality of organisational processes with a patient-centred focus, utilising objective information (real-time data).<br>CQI uses a structured "form": a set of standard tools that are used flexibly by CQI mentors. These tools are allowed to function differently in different contexts. The tools include:<br>• Process maps [36]: to understand existing clinic systems by documenting clinic workflow on specific activities (e.g., VL monitoring). Process maps serve to target improvement activities.<br>• Fishbone diagrams [37]: to identify root causes of a target outcome, by prespecified categories (e.g., patient level, clinic level, health worker level).<br>• PDSA cycles [38]: to test process improvements. These iterative action cycles start with a specific aim and a well-defined process change activity. The process change is adopted, adapted, or abandoned based on findings of the review phase ("Study") of the cycle.<br>• Run charts [39]: to monitor change in processes, outputs and outcomes. Run charts are plotted during improvement activities using clinic-level data, monitor target outcome time trends (e.g., VL monitoring), and provide visual feedback on trend changes due to the intervention.<br>**Hard copies of the following documents were also used:**<br>• South African eMTCT guidelines at each clinic (provided by the DoH)<br>• DoH eMTCT clinical care monitoring forms<br>• DoH monthly tally sheets (for monthly collation of DoH M&E indicators)<br>• DoH routine M&E clinic registers: to identify ANC patients eligible for VL monitoring and repeat HIV testing<br>**Other items required:**<br>• A projector for presentations<br>• Stationery for using CQI tools and note taking<br>• Refreshments for CQI introductory and action learning meetings to encourage and thank staff |
|---|---|
| What, procedures (4); when and how much (8) | The CQI intervention targeted health workers at clinics. It consisted of 2 phases: an intensive phase and a maintenance phase.<br>**Intensive**<br>*Initial induction visits*:<br>• During 2 weeks leading up to the scheduled rollover date for the intervention<br>• Up to 5 visits in each clinic: each visit was planned to last 1–4 hours except for the very first visit (which was planned to last a whole day)<br>• <u>Activities</u>: introduce the team and initiate relationships, introduce CQI, undertake situational analysis (understand resources and existing clinical processes), baseline data analysis and review quality of clinic data sources (e.g., registers)<br>*Intervention visits*:<br>• During the 2-month intervention step<br>• Up to 7 intervention visits: each visit was planned to last 1–4 hours.<br>• <u>Activities</u>: use CQI tools to identify clinic-specific gaps in implementing eMTCT guidelines and brainstorm solutions<br>*Support visits*:<br>• During the 2-month intervention step: each visit was planned to last 1–2 hours<br>• Up to 12 support visits in each clinic during the intensive phase<br>• <u>Activities</u>: provide ongoing support on progress and review of PDSA cycles<br>*Action learning sessions*:<br>• Held at the end of each 2-month intervention step: each visit was planned to last a whole day<br>• <u>Activities</u>: consolidate learning and facilitate buy-in of the next randomised clinic<br>**Maintenance**<br>*Support visits or maintenance visits*:<br>• Monthly after the 2-month intervention step: each visit was planned to last 1–2 hours<br>• Varied total number of visits depending on intervention rollover date—up to 12 visits for the first randomised clinic<br>• <u>Activities</u>: provide ongoing support on progress, review of PDSA cycles, ongoing mentorship to resolve problems and support to improve other relevant aspects of clinical care (e.g., infant HIV PCR testing) if time permitted |
| Who provided (5) | The CQI intervention was delivered in person by the UKZN CRH CQI mentors who travelled from Durban, KwaZulu-Natal, to the study community. The mentors had previous experience in a large CQI project on prevention of MTCT elsewhere in South Africa.<br>The CRH mentors consisted of 2 isiZulu-speaking South African nurses and a data capturer, with close support from an improvement advisor (a consultant obstetrician with CQI experience), scientific advisor, and data manager.<br>Apart from study objectives and study design, no additional training was provided to the CRH team. Their competence in delivering the intervention was not formally assessed. |
| How (6) | The CRH mentors identified a team of clinic-based health workers willing to participate in CQI activities (clinic CQI team). The clinic operational managers selected health workers employed in areas relevant to the intervention. The CRH mentors aimed to recruit health workers able to commit to group CQI activities for most of the study period to facilitate sustainability. |

(*Continued*)

**Table 1.** (Continued)

| | |
|---|---|
| **Where (7)** | Clinic visits were conducted in a meeting room at each clinic.<br>Action learning sessions were held at larger venues, including the AHRI auditorium, to accommodate the number of participants. |
| **Tailoring (9)** | Process changes to be implemented varied by clinic depending on findings of root-cause analyses. All standard CQI tools were implemented at all clinics.<br>The time constraints imposed by the study design required a prioritisation of intervention components: where the situational analysis identified multiple systemic gaps, some components such as M&E data quality could not be addressed as rigorously as others. The CRH improvement advisor supported and guided this prioritisation. |
| **Modification (10)** | No modifications were made. |
| **How well, planned (11)** | A schedule of visits was planned with a standard 'dose' for each intervention step, prior to the first scheduled clinic rollover. We assessed intervention adherence by comparing the planned schedule of visits (and type) against CRH mentor reports submitted at the end of each intervention step. The reports contained information on each clinic as follows:<br>• Actual visit dates, visit type (items (4) and (8) above), and clinic CQI team attendees<br>• Process maps<br>• Situational analysis findings, including barriers to VL monitoring and repeat HIV testing<br>• PDSA cycles implemented and outcomes<br>• Successes and challenges implementing CQI |
| **How well, actual (12)** | 'Dose': The overall 'dose' was greater than planned at the outset. Actual visit summaries compared with scheduled visits will be reported in a separate process evaluation.<br>'Reach': The health workers at all 7 facilities were enthusiastic about CQI.<br>Fidelity: The situational analyses identified staffing shortages as an important limitation in all clinics—some staff could not consistently attend CQI meetings or dedicate time to CQI implementation. DoH redeployment of lay counsellors during the study period may have affected HIV testing practices. Almost all clinics had no formal patient tracking system to identify women eligible for HIV care tests prior to CQI intervention rollover. |

**Abbreviations:** AHRI, Africa Health Research Institute; ANC, antenatal care; CQI, continuous quality improvement; CRH, Centre for Rural Health; DoH, South African National Department of Health; eMTCT, elimination of mother-to-child transmission of HIV; HIV PCR, polymerase chain reaction (nucleic acid amplification test for detecting HIV infection); M&E, monitoring and evaluation; MONARCH, Management and Optimisation of Nutrition, Antenatal, Reproductive, Child health & HIV care; MTCT, mother-to-child transmission of HIV; NPT, Normalisation Process Theory; PDSA, Plan-Do-Study-Act; TIDieR, Template for Intervention Description and Replication; TQM, Total Quality Management; UKZN, University of KwaZulu-Natal; VL, HIV viral load

with clearly defined goals and activities and were implemented flexibly based on need (Table 1). Patient care pathways in the clinic were documented using process maps [36] to identify areas for improvement (e.g., filing of VL results). Barriers and enablers of target endpoints (e.g., VL monitoring) were identified with fishbone diagrams [37], providing the opportunity for comprehensive, clinic-wide improvement. Improvement activities were reviewed using iterative Plan-Do-Study-Act (PDSA) cycles during which "one learns from taking action" in real time (tests of change), unlike awaiting the results of a formal research study [38]. Run charts—outcome time trends plotted during the course of improvement activities—provided visual feedback on whether any changes were likely due to the intervention [39]. As part of the IHI breakthrough collaborative model, the CRH CQI mentors also conducted action learning sessions to consolidate learning, share experiences between healthcare facilities, and motivate collaboration.

The intervention delivery according to the stepped-wedge study design is described in Table 1. The CRH team delivered CQI intensively to each cluster during the 2-month intervention step and then continued with the less intensive intervention during the maintenance phase (Table 1, Fig 2). They delivered a standard "dose" of approximately 19 visits during the intervention phase (2–3 visits per week) and continued with approximately monthly visits during the maintenance phase (Fig 2) for ongoing support and mentorship. The CRH mentors held action learning sessions at the end of each intervention step. In typical CQI delivery, health workers from all clinics in an intervention community would concurrently engage in the intervention and attend all action learning sessions. In our CQI delivery, health workers

participated in the intervention and the action learning sessions only during the phase when their clinic was in the intervention arm of our trial. Table 1 further describes materials; procedures; how and where the intervention was delivered (including duration and timing of CQI visits); and how we measured "dose," "reach," and fidelity of the intervention [41].

## Comparator

During control steps of the study design, health workers continued providing antenatal and postnatal care as usually implemented within routinely available resources.

## Participants

**Clusters.** Clusters were defined as described earlier. ANC health workers in clusters participated in CQI based on availability and ability to commit to CQI ideally for the entire study period. The CQI mentors tried to recruit health workers in leadership roles (e.g., operational managers, professional nurses) to clinic CQI teams to increase the likelihood that CQI activities would continue after the end of the intervention.

**Individuals.** For the primary endpoints, all women aged ≥18 years were eligible for recruitment at delivery if they were resident in the PIPSA area during pregnancy and/or had ever attended any of the 7 study clinics for ANC in pregnancy.

## Randomisation and blinding

We have described our randomisation procedure in detail elsewhere [30]. Briefly, the unit of randomisation was cluster balanced by patient volume. A senior biostatistician external to the study team performed the randomisation of all clusters during the baseline and before the first intervention step. Investigators and healthcare workers in the clusters were blinded to randomisation until the AHRI Chief Information Officer revealed each randomised cluster to the AHRI study team 2 weeks prior to the scheduled intervention rollover date for each cluster.

## Procedures

**Endpoints.** The pre-specified registered primary endpoints were indicators of quality of care in HIV-related ANC: (i) VL monitoring among pregnant women living with HIV and (ii) repeat HIV testing among pregnant women not living with HIV. We report the intervention impact on both primary endpoints in this manuscript. We will analyse and report the secondary endpoints elsewhere.

**Data sources.** The data on our primary endpoints were sourced entirely from routine patient medical records (maternity case records) [30], which were photographed at delivery. All clusters provided pre-CQI, CQI implementation, and post-CQI data continuously throughout the study. As the maternity case records were first accessed after delivery, ANC data were captured retrospectively—this extended the baseline observation period by an additional 6 months, resulting in a total data collection period of 19 months and a total observation period of 25 months. The period after all clusters had received the CQI intervention was 4.5 months (Fig 2). We used a Research Electronic Data Capture (REDCap) study database for data entry [42]. We collected outcome data continuously over the study at all 7 primary care clinics participating in this study. We also collected outcome data at Hlabisa Hospital maternity ward, because most women living in the study subdistrict deliver at this hospital.

We also conducted a process evaluation to better understand intervention delivery and explain our primary findings. For this, we sourced field notes and reports by the CRH team collated every 2 months. The reports described actual visit dates and type, results of the root-

cause analyses, the improvement interventions (including PDSA cycles), successes and challenges, as well as other observations, including impressions of health worker receptivity to CQI (Table 1). We further conducted semi-structured interviews with consenting health workers on their experiences of implementing CQI. We describe in detail the methods, data, and results of the process evaluation in an upcoming scientific publication.

## Statistical methods

**Power calculation and sample size.** As we describe in our protocol paper [30], we assumed for our baseline power calculation—informed by local routine data—that without CQI 40% of all pregnant women living with HIV would receive a test for VL monitoring and 65% of pregnant women not living with HIV would receive a repeat HIV test. We further assumed that half of all pregnant women would be HIV positive and that pregnant women would make 3 ANC visits. We assumed an intracluster correlation coefficient (ICC) of 0.10, which is a conservative assumption compared to other ICCs measured in similar settings [43]. We assumed missing data from 15% of enrolled women. If we enrolled a total of 1,260 pregnant women (i.e., 630 women living with HIV and 630 women not living with HIV), we estimated 80% power to detect at least a 15-percentage-point increase in our 2 primary endpoints at the 5% significance level [30]. In discussion with local stakeholders, we identified this minimum detectable difference over the course of a pregnancy as relevant for health policy and clinical practice.

**Statistical analyses.** We performed intention-to-treat (ITT) analyses based on the clinic attended at the first antenatal booking visit—individuals declared their "intention" to attend that same facility for the remainder of pregnancy. Although it is well established that the AHRI surveillance population is mobile [31], ITT assumes exposure to a single clinic for the entire duration of pregnancy regardless of actual attendance elsewhere. All participants were assigned CQI exposure status at each ANC visit (by the actual date of that visit) according to the exposure status of their clinic at that time. Participants whose assigned clinic rolled over to CQI during their ANC thus had 1 or more initial ANC visits that were CQI unexposed and 1 or more later visits that were CQI exposed. The beginning of each step (CQI rollover date) was defined as the date of the first actual CQI intervention visit in the randomised cluster.

The binary VL monitoring endpoint was measured in pregnant women living with HIV and defined as a documented VL test performed at a particular visit. Each ANC visit was eligible for an endpoint assessment on or after the first documented HIV-positive status irrespective of whether ART was initiated or continued in pregnancy and irrespective of actual VL results. This definition accounts for real-life imperfections in adherence to guidelines and documentation of ART prescriptions. Women who seroconverted from HIV-negative to HIV-positive status during the study were not included in the analysis of the VL monitoring endpoint.

The binary repeat HIV testing endpoint was measured in pregnant women not living with HIV and defined as a subsequent documented HIV test at a particular visit. Each ANC visit following the first documented negative HIV test was eligible for assessment of this endpoint. ANC visits among women who subsequently tested HIV positive were not eligible for the repeat HIV testing endpoint after the first documented HIV-positive test. We did not restrict our endpoint definitions by visit number or gestation, to allow for real-life imperfections in adherence to guidelines.

**Primary analyses.** The 2018 extension of the CONSORT statement for stepped-wedge cluster RCTs states that "in addition to reporting a relative measure of the effect of the intervention, it can be helpful to report an absolute measure of the effect" [33]. The reason for these

recommendations are that "relative measures of the effects are often more stable across different populations" [33]. Relative measures are therefore more useful than absolute measures for policy makers considering transferring an intervention from one context to another one. In contrast, "absolute measures of effects are more easily understood" [33]. We follow these recommendations and report and interpret both the relative and absolute effect sizes as our primary analyses.

For the primary analysis, we estimated the relative effect sizes using modified Poisson mixed effects generalised linear regression models. Modified Poisson regression has become a standard for the analysis of RCTs with binary outcomes [44,45] because it has advantages over logistic and log binomial regression models [44,46]. One advantage of modified Poisson regression is that it directly generates risk ratios rather than the odds ratios that logistic regression generates. Risk ratios are easier to interpret than odds ratios and, unlike odds ratios, are collapsible [47–49].

Another advantage of modified Poisson regression is that it does not suffer from the convergence problems that commonly arise with another approach to estimating risk ratios, log-binomial regression [50–52]. In addition, modified Poisson regression models are more robust to model misspecification than log-binomial regression models [53]. A disadvantage of modified Poisson regression is that it can produce predicted probabilities greater than unity. However, several simulation studies have shown that modified Poisson regression provides risk ratio estimates equivalent to regression models that cannot produce such predicted probabilities, such as log-binomial regression [50,54–56]. Modified Poisson regression models are thus a good choice for estimating risk ratios in RCTs [45]. To measure absolute effect sizes, we used mixed effects linear probability regression models.

We used 3 models to estimate the effect of the CQI intervention. First, we used the standard Hussey and Hughes model, which includes time-step fixed effects and clinic random effects (Model 1) [57]. Second, we used an extension to this model with a nested random clinic–time step interaction term (Model 2). This extension is recommended by Hemming and colleagues [58], because it allows secular time trends to vary randomly by clinic. Third, we extended the standard Hussey and Hughes model with individual random effects nested within clinic random effects (Model 3). This model accounts not only for clustering of outcomes by clinic but also for clustering of outcomes within individual women across time steps. In all models, we used cluster robust standard errors to further adjust for clustering and model misspecification [44,45].

In addition to the per-visit effect sizes described earlier, we also computed the cumulative absolute probabilities of attaining our endpoints. For this purpose, we used the per-visit absolute effect sizes measured in each model and applied the exponential formula to estimate the cumulative probabilities across the median number of visits during a pregnancy [59].

**Sensitivity analyses.** For both of our primary analyses—estimating relative and absolute risk—we also ran regressions with additional control variables. We added (i) maternal age and parity; (ii) gestation at each ANC visit and total number of ANC visits attended in pregnancy; and (iii) maternal age, parity, gestation at each ANC visit, and total number of ANC visits.

**Analysis of effect heterogeneity.** We measured CQI effect heterogeneity by duration of CQI exposure, using the same mixed effects regression models as in the primary analyses but replacing the fixed effect for overall CQI exposure with fixed effects for CQI exposure for each time step since rollover to CQI.

We used Stata version 15.0 (StataCorp LLC, College Station, TX) for all statistical analyses.

Ethics approval for the study was obtained from the University of KwaZulu-Natal Biomedical Research Ethics Committee (reference BE209/14). The ethics approval included a waiver of the requirement for individual consent to access routine clinical data from maternity case

records, excluding labour and delivery clinical notes. Engagement meetings were held with subdistrict and district-level DoH partners prior to study commencement to share our study objectives and introduce the intervention. Standard DoH approvals for commencing the study were also obtained as part of a Memorandum of Understanding between AHRI and the DoH. Following the analyses of our results and prior to the publication of this paper, we held engagement workshops with the sub-district and district-level DoH partners, as well as with the primary funders of this study (the Delegation of the European Commission to South Africa). During these workshops, we jointly interpreted our findings and derived policy recommendations.

Although this is a low-risk health systems implementation trial, an independent Data Safety and Monitoring Board (DSMB) annually reviewed study progress. No adverse event data were formally collected, because the intervention targeted health workers in clinical facilities.

## Results

All 7 primary care clinics and Hlabisa Hospital maternity ward agreed to participate in the study. Maternity case records from 2,498 women who delivered between 15 July 2015 and 30 January 2017 were accessed. The clinic nurses and local DoH staff were continuous partners in this intervention throughout the study duration. No adverse events, complaints, or other deleterious effects of the CQI intervention were reported. The actual intervention rollover dates of clusters are shown in S1 Table.

The health workers appreciated the CQI intervention. For instance, the CRH team observed that the clinic health workers who participated in the CQI intervention were enthusiastic about their increased capacity to improve quality of services. Several clinic health workers expressed an interest in further training and mentoring on quality improvement tools and approaches. Finally, clinic health workers urged the CRH team to continue to work with them on quality improvement beyond the end of this study. We will report further details on these findings in an upcoming publication on the process evaluation that accompanied this trial. Preliminary results of situational analyses (conducted prior to intervention rollover) identified staffing shortages as well as lack of formal tracking systems for patients—and for test results— as potential impediments to the intervention activities described in Table 1.

Most maternity case records contained ANC clinical information. Among the 2,498 women recruited to this study (Fig 3), 5 had blank maternity case records because their previous records had been lost, and a further 19 had blank records because they never attended ANC. Of all women recorded in our study database, 2,160 participants were eligible for our primary ITT analysis (contributing 13,212 observations) during pregnancy: 338 women were excluded from our primary analysis because their HIV status at delivery was unknown, their first ANC clinic was not a study clinic, or their first ANC clinic was unknown (Fig 3).

We estimated the total number of women living with HIV who attended a study clinic at their first ANC visit to be 1,026 (1,011 with documented HIV status plus 15 women with unknown HIV status by delivery), of whom 99% were diagnosed HIV positive—and could therefore start ART.

Median age at first booking was 25 (interquartile range [IQR] 21 to 30), and median gestation at first booking was 19 weeks (IQR 15 to 24). Median parity was 1 (IQR 0 to 2). Median number of antenatal visits was 6 (IQR 4 to 8), and 86.9% of women (95% CI 82.5% to 90.3%) attended at least 4 ANC visits during pregnancy. Around half the women (50.3%, 95% CI 45.1% to 55.4%) attended their first ANC visit before 20 weeks' gestation. HIV prevalence at first ANC visit was 47.4% (95% CI 42.3% to 52.7%) and varied by age group (S1 Fig). There were no major differences in individual participant baseline variables by arm (Table 2), i.e., the

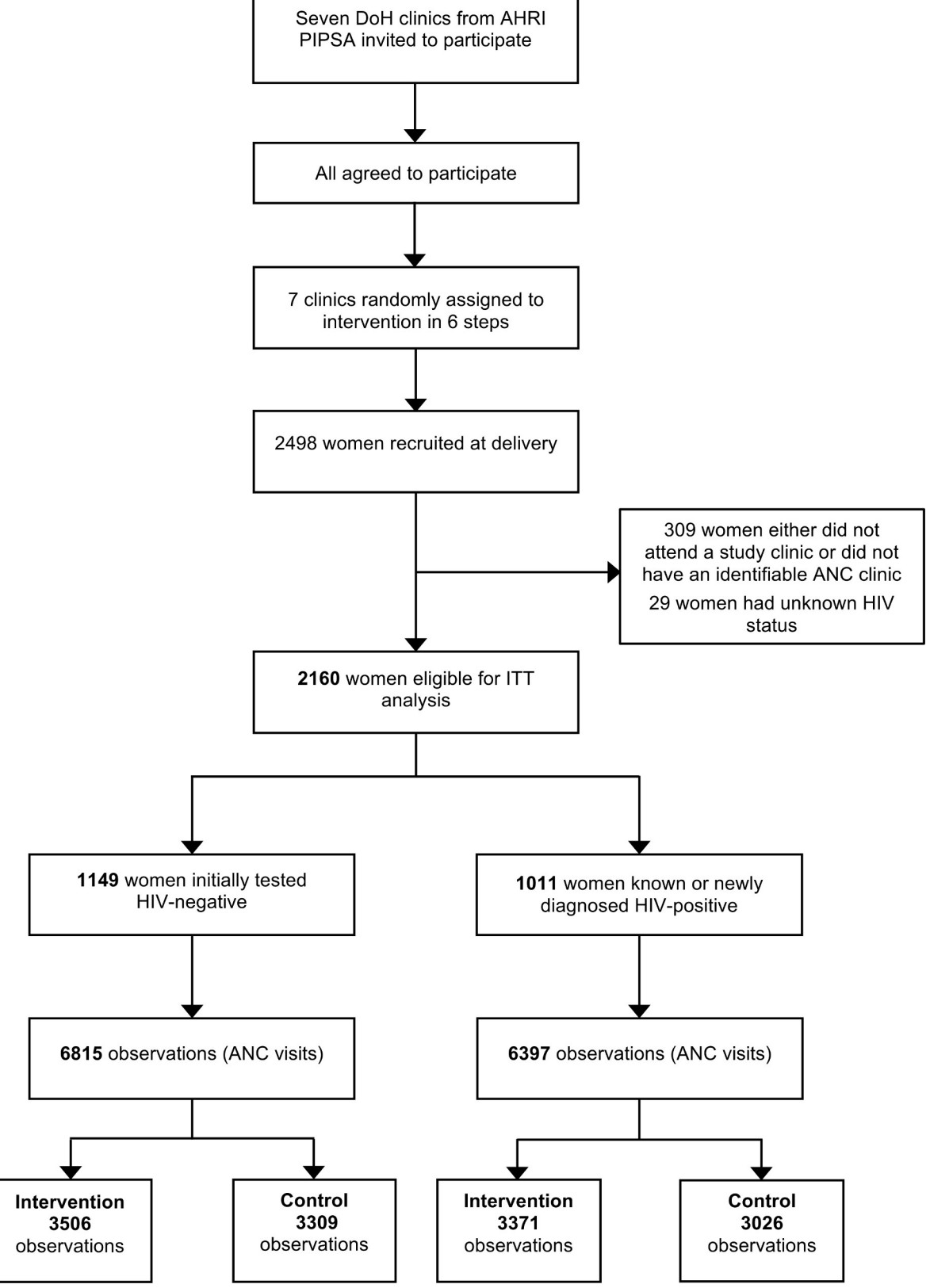

**Fig 3. Participant flow diagram.** Of 2,160 participants, women were assigned for analysis of each endpoint based on their first documented HIV status; 1,011 women who were HIV positive at first documented HIV status were analysed for the VL monitoring endpoint; 1,149

women with a negative HIV test at first documented HIV status were analysed for the repeat HIV testing endpoint and included 12 women who subsequently seroconverted to HIV-positive status (up to and including the date of seroconversion). AHRI, Africa Health Research Institute; ANC, antenatal care; DoH, South African National Department of Health; ITT, intention-to-treat; PIPSA, AHRI Population Intervention Platform Surveillance Area; VL, viral load.

2 arms were balanced on observed baseline characteristics. Among all eligible participants 822 of 2,160 (38.1%) were never exposed to CQI during their pregnancy (i.e., during all of their ANC clinic visits the clinic[s] had not yet received the CQI intervention), 870 of 2,160 (40.3%) were fully exposed to CQI (i.e., during all of their ANC clinic visits the clinic[s] had already received the CQI intervention), and the remaining participants were partially exposed (i.e., during initial ANC clinic visits the clinic[s] had not yet received the CQI intervention, while during later ANC clinic visits the clinic[s] had received the intervention).

**Table 2. Summary of participant characteristics by CQI intervention exposure status.**

| Characteristic | Intervention $n$ = 1,154 (53.4%)[a] | Control $n$ = 1,006 (46.6%)[a] |
|---|---|---|
| **Clinic[b,c]** | | |
| Clinic 1 | 229 (19.8%) | 84 (8.4%) |
| Clinic 2 | 456 (39.5%) | 243 (24.2%) |
| Clinic 3a | 53 (4.6%) | 58 (5.8%) |
| Clinic 3b | 19 (1.6%) | 23 (2.3%) |
| Clinic 4 | 302 (26.2%) | 326 (32.4%) |
| Clinic 5 | 45 (3.9%) | 82 (8.2%) |
| Clinic 6 | 50 (4.3%) | 190 (18.9%) |
| **Age group (years)[b]** | | |
| <20 | 178 (15.4%) | 175 (17.4%) |
| 20–24 | 389 (33.7%) | 298 (29.6%) |
| 25–29 | 276 (23.9%) | 262 (26.0%) |
| 30–34 | 194 (16.8%) | 173 (17.2%) |
| 35–39 | 98 (8.5%) | 73 (7.3%) |
| 40+ | 19 (1.6%) | 24 (2.4%) |
| Missing values | 0 | 1 (0.1%) |
| **HIV status at first ANC visit[b]** | | |
| Positive | 549 (47.6%) | 455 (45.2%) |
| Negative | 585 (50.7%) | 527 (52.4%) |
| Unknown | 20 (1.7%) | 24 (2.4%) |
| **Gestation at first ANC visit[b]** | | |
| ≤12 weeks | 135 (11.7%) | 104 (10.3%) |
| 13–19 weeks | 375 (32.5%) | 290 (28.8%) |
| 20–27 weeks | 351 (30.4%) | 315 (31.3%) |
| ≥28 weeks | 133 (11.5%) | 96 (9.5%) |
| Missing values | 160 (13.9%) | 201 (20.0%) |

[a]For this summary table, women with partial exposure to CQI are allocated to either the intervention or control arm based on majority of pregnancy spent exposed or unexposed.

[b]Percentages may not add up to 100% due to rounding.

[c]Clinics are listed in order of rollover to the intervention. Clinics 3a and 3b, the two smallest clinics, formed a single intervention cluster. Due to the stepped-wedge design, clinics rolling over later in the study contributed fewer participants overall exposed to the intervention.

**Abbreviations:** ANC, antenatal care; CQI, continuous quality improvement

## Pregnant women living with HIV

We analysed 1,011 women whose first documented HIV status was HIV positive. At any time in pregnancy, 93.8% of women living with HIV had a documented ART prescription, and 56.3% had received a VL test. Among those women who had received a VL test, only 52.4% had a documented result of which 85.2% were <200 copies/mL (Table 3). ART prescriptions, VL monitoring, and suppression in late pregnancy are presented in Table 3.

## Pregnant women not living with HIV

Among the 1,149 women with an initial negative HIV test, 768 of 1,149 (66.8%) had at least one repeat HIV screen during pregnancy. Repeat HIV testing within 3 months prior to delivery is presented in Table 3. Twelve women (1.0%) seroconverted during pregnancy.

**Table 3. Descriptive outcomes aggregated across all participants.**

| Outcome | Proportion (95% CI) |
|---|---|
| Women who were HIV positive (n = 1,011) | |
| ART prescription ever in pregnancy | 948/1,011 (93.8%) (88.7% to 96.7%) |
| ART prescription within 1 month[a] before delivery | 570/1,011 (56.4%) (41.2% to 70.4%) |
| VL performed ever in pregnancy | 569/1,011 (56.3%) (49.6% to 62.8%) |
| VL performed within 3 months[b] before delivery | 403/1,011 (39.9%) (34.6% to 45.4%) |
| VL results documented ever | 298/569 (52.4%) (35.8% to 68.5%) |
| VL results documented within 3 months before delivery | 155/403 (38.5%) (26.8% to 51.6%) |
| VL suppressed ever | |
| <200 copies/mL | 254/298 (85.2%) (78.1% to 90.3%) |
| <1,000 copies/mL | 277/298 (93.0%) (88.2% to 95.9%) |
| VL suppressed within 3 months before delivery | |
| <200 copies/mL | 132/155 (85.2%) (81.4% to 88.3%) |
| <1,000 copies/mL | 143/155 (92.3%) (85.2% to 96.1%) |
| Women with an initial negative HIV test (n = 1,149) | |
| Repeat HIV test ever in pregnancy | 768/1,149 (66.8%) (58.7% to 74.1%) |
| Repeat HIV test within 3 months[b] before delivery | 730/1,149 (63.5%) (55.6% to 70.8%) |
| Seroconverted to HIV positive | 12/1,149 (1.0%) (0.5% to 2.0%) |

[a]ART prescriptions within 1 month prior to delivery rather than 3 months are presented, because the ART prescription frequency is generally 1 month in duration at facilities.

[b]VL monitoring and repeat HIV testing within 3 months prior to delivery are presented, because the guidelines-recommended testing frequency is, on average, every 3 months. This measure also describes proximity of testing to the delivery date given the importance of early HIV diagnosis, ensuring VL suppression in the peripartum period and appropriate infant prophylaxis.

**Abbreviations:** ART, antiretroviral therapy; VL, HIV viral load

## Effect of CQI on HIV care tests

**Primary analyses—Pregnant women living with HIV.** Women exposed to CQI were significantly more likely to receive a VL test: the relative effect sizes (relative risks [RRs]) in our 3 primary analyses ranged from 1.38 to 1.39 (Fig 4, Table 4) and were all highly significant (all $p < 0.001$). The absolute effect sizes of receiving a VL test per ANC visit in our 3 primary analyses ranged from 3.9 to 4.1 percentage points higher (Table 5) in the intervention compared to the control arm. With a median of 6 ANC visits per pregnancy in this study, the absolute effect size per visit translates to a cumulative probability of receiving a VL test in pregnancy of 58% in the intervention arm compared to 46% in the control arm. With the recent WHO recommendations for at least 8 ANC visits [60], these values are equivalent to a cumulative probability of receiving a VL test of 69% in the intervention arm compared to 57% in the control arm.

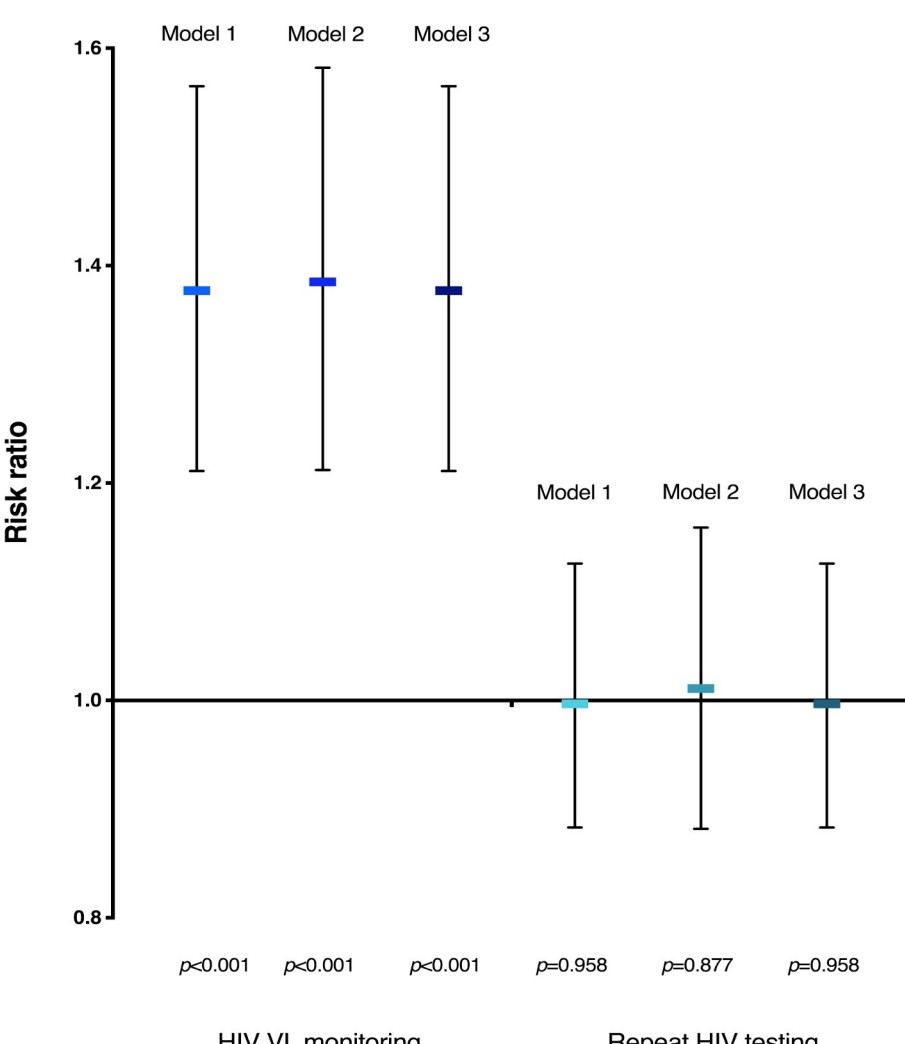

**Fig 4. Effects of CQI on VL monitoring and repeat HIV testing.** Model 1 includes time-step fixed effects and clinic random effects. Model 2 includes time-step fixed effects, clinic random effects, and a random clinic–time step interaction term. Model 3 includes time-step fixed effects, clinic random effects, and individual random effects. CQI, continuous quality improvement; VL, HIV viral load

**Table 4. Regression models for HIV VL monitoring and repeat HIV testing: RR.**

| Variable | VL monitoring $n = 6,397$ | | | Repeat HIV testing $n = 6,815$ | | |
|---|---|---|---|---|---|---|
| | Model 1 RR (95% CI) *p*-Value | Model 2 RR (95% CI) *p*-Value | Model 3 RR (95% CI) *p*-Value | Model 1 RR (95% CI) *p*-Value | Model 2 RR (95% CI) *p*-Value | Model 3 RR (95% CI) *p*-Value |
| **CQI effect** | | | | | | |
| | 1.38 (1.21 to 1.57) $p < 0.001$ | 1.39 (1.21 to 1.58) $p < 0.001$ | 1.38 (1.21 to 1.57) $p < 0.001$ | 1.00 (0.88 to 1.13) $p = 0.958$ | 1.01 (0.88 to 1.16) $p = 0.877$ | 1.00 (0.88 to 1.13) $p = 0.958$ |
| **Time step** | | | | | | |
| Step 0 | (base) | (base) | (base) | (base) | (base) | (base) |
| Step 1 | 1.37 (1.03 to 1.82) $p = 0.033$ | 1.34 (1.00 to 1.80) $p = 0.046$ | 1.37 (1.03 to 1.82) $p = 0.033$ | 0.76 (0.55 to 1.06) $p = 0.110$ | 0.74 (0.54 to 1.01) $p = 0.056$ | 0.76 (0.55 to 1.06) $p = 0.110$ |
| Step 2 | 1.26 (0.71 to 2.22) $p = 0.428$ | 1.24 (0.72 to 2.11) $p = 0.439$ | 1.26 (0.71 to 2.22) $p = 0.428$ | 1.12 (0.81 to 1.55) $p = 0.510$ | 1.05 (0.74 to 1.49) $p = 0.781$ | 1.12 (0.81 to 1.55) $p = 0.510$ |
| Step 3 | 1.38 (0.92 to 2.05) $p = 0.115$ | 1.34 (0.87 to 2.07) $p = 0.190$ | 1.38 (0.92 to 2.05) $p = 0.115$ | 1.24 (0.78 to 1.96) $p = 0.357$ | 1.16 (0.76 to 1.78) $p = 0.489$ | 1.24 (0.78 to 1.96) $p = 0.357$ |
| Step 4 | 1.38 (0.90 to 2.12) $p = 0.141$ | 1.34 (0.84 to 2.13) $p = 0.225$ | 1.38 (0.90 to 2.12) $p = 0.141$ | 1.32 (0.89 to 1.94) $p = 0.167$ | 1.27 (0.83 to 1.95) $p = 0.276$ | 1.32 (0.89 to 1.94) $p = 0.167$ |
| Step 5 | 1.58 (0.99 to 2.51) $p = 0.055$ | 1.56 (0.99 to 2.45) $p = 0.056$ | 1.58 (0.99 to 2.51) $p = 0.055$ | 1.24 (1.01 to 1.53) $p = 0.038$ | 1.18 (0.93 to 1.51) $p = 0.169$ | 1.24 (1.01 to 1.53) $p = 0.038$ |
| Step 6 | 1.34 (0.85 to 2.12) $p = 0.203$ | 1.33 (0.87 to 2.05) $p = 0.192$ | 1.34 (0.85 to 2.12) $p = 0.203$ | 1.40 (1.01 to 1.92) $p = 0.042$ | 1.34 (0.95 to 1.89) $p = 0.100$ | 1.40 (1.01 to 1.92) $p = 0.042$ |
| Step 7 | 1.09 (0.60 to 1.99) $p = 0.784$ | 1.07 (0.59 to 1.94) $p = 0.829$ | 1.09 (0.60 to 1.99) $p = 0.784$ | 2.47 (1.53 to 4.01) $p < 0.001$ | 2.21 (1.35 to 3.60) $p = 0.002$ | 2.47 (1.53 to 4.01) $p < 0.001$ |
| **Random effects (95% CI)** | | | | | | |
| Clinic | 0.0098 (0.0003 to 0.3471) | 0.0080 (0.0001 to 0.6180) | 0.0098 (0.0003 to 0.3471) | $1.13 \times 10^{-33}$ ($6.10 \times 10^{-34}$ to $2.09 \times 10^{-33}$) | $6.11 \times 10^{-30}$ ($6.13 \times 10^{-85}$ to $6.10 \times 10^{+25}$) | $1.13 \times 10^{-34}$ ($5.39 \times 10^{-36}$ to $2.38 \times 10^{-33}$) |
| Clinic–time | N/A | 0.0111 (0.0003 to 0.3683) | N/A | N/A | 0.0321 (0.0176 to 0.0587) | N/A |
| Individual | N/A | N/A | $9.54 \times 10^{-35}$ ($7.54 \times 10^{-35}$ to $1.21 \times 10^{-34}$) | N/A | N/A | $1.49 \times 10^{-33}$ ($1.40 \times 10^{-33}$ to $1.58 \times 10^{-33}$) |
| **Log pseudolikelihood** | −2,282.72 | −2,282.40 | −2,282.72 | −2,425.31 | −2,421.91 | −2,425.31 |
| **Akaike information criterion** | 4,577.44 | 4,576.80 | 4,577.44 | 4,862.62 | 4,857.82 | 4,862.62 |

Model 1 includes time-step fixed effects, clinic random effects, and cluster robust standard errors.

Model 2 includes time-step fixed effects, clinic random effects, a nested random clinic-time step interaction effect, and cluster robust standard errors.

Model 3 includes time-step fixed effects, clinic random effects, nested individual random effects, and cluster robust standard errors.

**Abbreviations:** CQI, continuous quality improvement; N/A, not applicable; RR, relative risk; VL, HIV viral load

**Primary analyses—Pregnant women not living with HIV.** CQI did not have a statistically significant effect on repeat HIV testing in any of our 3 primary analyses. The RRs ranged from 1.00 to 1.01, and the corresponding *p*-values ranged from 0.877 to 0.958 (Table 4, Fig 4).

**Table 5. Regression models for HIV VL monitoring and repeat HIV testing estimating absolute risk difference.**

| Variable | VL monitoring n = 6,397 | | | Repeat HIV testing n = 6,815 | | |
|---|---|---|---|---|---|---|
| | Model 1 | Model 2 | Model 3 | Model 1 | Model 2 | Model 3 |
| | Coefficient (95% CI) p-Value | Coefficient (95% CI) p-Value | Coefficient (95% CI) p-Value | Coefficient (95% CI) p-Value | Coefficient (95% CI) p-Value | Coefficient (95% CI) p-Value |
| **CQI effect** | | | | | | |
| | 0.039 (0.022 to 0.055) $p < 0.001$ | 0.041 (0.021 to 0.061) $p < 0.001$ | 0.039 (0.022 to 0.055) $p < 0.001$ | −0.000 (−0.017 to 0.016) $p = 0.958$ | 0.003 (−0.018 to 0.025) $p = 0.758$ | −0.000 (−0.017 to 0.016) $p = 0.958$ |
| **Time step** | | | | | | |
| Step 0 | (base) | (base) | (base) | (base) | (base) | (base) |
| Step 1 | 0.027 (0.004 to 0.050) $p = 0.020$ | 0.024 (−0.002 to 0.050) $p = 0.070$ | 0.027 (0.004 to 0.050) $p = 0.020$ | −0.028 (−0.062 to 0.005) $p = 0.096$ | −0.036 (−0.069 to −0.002) $p = 0.035$ | −0.028 (−0.062 to 0.005) $p = 0.096$ |
| Step 2 | 0.018 (−0.036 to 0.072) $p = 0.518$ | 0.013 (−0.038 to 0.065) $p = 0.615$ | 0.018 (−0.036 to 0.072) $p = 0.518$ | 0.014 (−0.028 to 0.056) $p = 0.512$ | 0.002 (−0.047 to 0.050) $p = 0.947$ | 0.014 (−0.028 to 0.056) $p = 0.512$ |
| Step 3 | 0.029 (−0.007 to 0.064) $p = 0.114$ | 0.024 (−0.020 to 0.068) $p = 0.287$ | 0.029 (−0.007 to 0.064) $p = 0.114$ | 0.029 (−0.035 to 0.093) $p = 0.376$ | 0.016 (−0.039 to 0.072) $p = 0.563$ | 0.029 (−0.035 to 0.093) $p = 0.376$ |
| Step 4 | 0.028 (−0.012 to 0.069) $p = 0.173$ | 0.022 (−0.028 to 0.073) $p = 0.387$ | 0.028 (−0.012 to 0.069) $p = 0.173$ | 0.038 (−0.016 to 0.092) $p = 0.167$ | 0.032 (−0.035 to 0.098) $p = 0.348$ | 0.038 (−0.016 to 0.092) $p = 0.167$ |
| Step 5 | 0.048 (0.002 to 0.093) $p = 0.040$ | 0.045 (−0.002 to 0.091) $p = 0.059$ | 0.048 (0.002 to 0.093) $p = 0.040$ | 0.029 (0.004 to 0.055) $p = 0.024$ | 0.019 (−0.019 to 0.057) $p = 0.321$ | 0.029 (0.004 to 0.055) $p = 0.024$ |
| Step 6 | 0.025 (−0.018 to 0.068) $p = 0.253$ | 0.022 (−0.019 to 0.063) $p = 0.294$ | 0.025 (−0.018 to 0.068) $p = 0.253$ | 0.047 (0.000 to 0.095) $p = 0.050$ | 0.039 (−0.017 to 0.094) $p = 0.172$ | 0.047 (0.000 to 0.095) $p = 0.050$ |
| Step 7 | −0.001 (−0.059 to 0.056) $p = 0.962$ | −0.006 (−0.066 to 0.054) $p = 0.853$ | −0.001 (−0.059 to 0.056) $p = 0.962$ | 0.177 (0.074 to 0.279) $p = 0.001$ | 0.149 (0.051 to 0.247) $p = 0.003$ | 0.177 (0.074 to 0.279) $p = 0.001$ |
| **Random effects (95% CI)** | | | | | | |
| Clinic | 0.0002 ($4.06 \times 10^{-06}$ to 0.0068) | 0.0001 ($1.60 \times 10^{-06}$ to 0.0133) | 0.0002 ($4.06 \times 10^{-06}$ to 0.0068) | $1.10 \times 10^{-34}$ ($6.78 \times 10^{-36}$ to $1.79 \times 10^{-33}$) | $7.44 \times 10^{-35}$ ($2.25 \times 10^{-38}$ to $2.45 \times 10^{-31}$) | $4.47 \times 10^{-35}$ ($1.79 \times 10^{-37}$ to $1.12 \times 10^{-32}$) |
| Clinic–time | N/A | 0.0002 (0.0000 to 0.0014) | N/A | N/A | 0.0014 (0.0007 to 0.0025) | N/A |
| Individual | N/A | N/A | $5.75 \times 10^{-36}$ ($5.78 \times 10^{-40}$ to $5.72 \times 10^{-32}$) | N/A | N/A | $2.34 \times 10^{-36}$ ($4.23 \times 10^{-37}$ to $1.29 \times 10^{-35}$) |
| **Log pseudolikelihood** | −1,711.95 | −1,711.22 | −1,711.95 | −2,178.47 | −2,169.58 | −2,178.47 |
| **Akaike information criterion** | 3,435.89 | 3,434.44 | 3,437.89 | 4,368.94 | 4,351.16 | 4,370.94 |

Model 1 includes time-step fixed effects, clinic random effects, and cluster robust standard errors.

Model 2 includes time-step fixed effects, clinic random effects, a nested random clinic-time step interaction effect, and cluster robust standard errors.

Model 3 includes time-step fixed effects, clinic random effects, individual random effects, and cluster robust standard errors.

**Abbreviations:** CQI, continuous quality improvement; N/A, not applicable; VL, HIV viral load

These numbers are equivalent to an absolute change in the probability of receiving a repeat HIV test per ANC visit of 0.0 to 0.3 percentage points (all $p \geq 0.758$, Table 5). With a median of 6 ANC visits in this study, this effect size per visit translates to a 59% to 60% cumulative probability of receiving a repeat HIV test during pregnancy in both the intervention and the

control arm. With the WHO-recommended 8 ANC visits, these numbers translate to 69% to 70% cumulative probability of receiving a repeat HIV test in both arms.

**Sensitivity analyses.** When we adjusted for additional covariates, all effect size estimates remained essentially the same as in the primary analyses. RRs for HIV VL monitoring ranged from 1.36 to 1.37 (all $p < 0.001$) with adjustments for maternal age and parity, 1.39 to 1.40 ($p$-values <0.001 to 0.001) with adjustments for gestation at each visit and total number of ANC visits, and 1.35 to 1.36 ($p$-values 0.005 to 0.009) with adjustments for all 4 covariates. The absolute effect sizes for VL monitoring per ANC visit were 3.7 to 4.0 percentage points higher (all $p < 0.001$) with adjustments for maternal age and parity, 4.4 to 4.6 percentage points (all $p < 0.001$) with adjustments for gestation at each visit and total number of ANC visits, and 4.2 to 4.4 percentage points (all $p \leq 0.008$) with adjustments for all 4 covariates.

RRs for repeat HIV testing ranged from 1.00 to 1.01 (all $p \geq 0.871$) with adjustments for maternal age and parity, 1.08 (all $p \geq 0.525$) with adjustments for gestation at each visit and total number of ANC visits, and 1.08 (all $p \geq 0.504$) with adjustments for all 4 covariates. The absolute effect sizes for repeat HIV testing per ANC visit ranged from 0.0 to 0.3 percentage points (all $p \geq 0.757$) with adjustments for maternal age and parity, 1.1 to 1.3 percentage points (all $p \geq 0.457$) with adjustments for gestation at each visit and total number of ANC visits, and 1.1 to 1.3 percentage points (all $p \geq 0.445$) with adjustments for all 4 covariates.

**CQI effect heterogeneity.** There was variability in CQI effect on VL monitoring with time since rollover to the intervention, in all 3 mixed effects models. Immediately after rolling over to the CQI intervention, the relative effect size was 1.17 ($p = 0.071$) in Model 1. After 2 months of CQI exposure (lag of 1 step) and 4 months (lag of 2 steps), the relative effect sizes were 1.42 ($p = 0.021$) and 1.45 ($p = 0.015$), respectively. There was no significant CQI effect beyond 6 months of exposure (Table 6). Similarly, the absolute effect sizes varied with the same durations of exposure to CQI in Model 1: 1.7 percentage points ($p = 0.070$) with immediate exposure, 4.4 percentage points ($p = 0.038$) after 2 months of exposure, 4.9 percentage points ($p = 0.035$) after 4 months of exposure, and no significant change beyond 6 months of exposure (Table 6). The results were essentially the same with Models 2 and 3 for relative and absolute effect sizes. There was no heterogeneity in CQI effect on repeat HIV testing by duration of CQI exposure in either the relative or absolute measures (Table 6).

## Discussion

Our stepped-wedge cluster RCT conducted under real-life conditions in rural South Africa showed that CQI improved antenatal HIV VL monitoring but did not improve repeat HIV testing. To our knowledge, this is the first RCT to show that CQI can improve quality of primary care in sub-Saharan Africa.

The significant improvement in the first of our 2 primary endpoints—VL monitoring—suggests that CQI may indeed be a good approach to improve quality of care in primary care. This finding is important for policy: CQI is an attractive approach for resource-poor settings because it works with local health workers and within existing resources. Moreover, CQI is already commonly used to improve quality of care in many resource-poor countries and communities, including sub-Saharan Africa [7–11]. Our trial provides evidence for these policies.

Both relative and absolute effect sizes are important for policy. We found a large relative effect of the CQI intervention on VL monitoring: CQI caused an almost 40% increase in VL monitoring. Relative effect sizes are commonly more stable across populations with different baseline attainments of an endpoint [61]. VL monitoring varies widely across ANC and HIV treatment programmes in sub-Saharan Africa [62,63]. Policy makers should therefore consider our relative effect size in the context of their health system's achievement in measuring this

**Table 6.  CQI effect heterogeneity by time since rollover to CQI.**

| Variable | VL monitoring $n = 6,397$ | | Repeat HIV testing $n = 6,815$ | |
|---|---|---|---|---|
| | Model 1 | Model 1 | Model 1 | Model 1 |
| | RR (95% CI) p-Value | Absolute RD (95% CI) p-Value | RR (95% CI) p-Value | Absolute RD (95% CI) p-Value |
| **CQI effect: Time steps since CQI rollover** | | | | |
| Immediate | 1.17 (0.99 to 1.39) $p = 0.071$ | 0.017 (−0.001 to 0.036) $p = 0.070$ | 1.05 (0.76 to 1.45) $p = 0.764$ | 0.007 (−0.040 to 0.055) $p = 0.766$ |
| 1 step | 1.42 (1.06 to 1.92) $p = 0.021$ | 0.044 (0.002 to 0.086) $p = 0.038$ | 1.04 (0.74 to 1.46) $p = 0.826$ | 0.007 (−0.050 to 0.064) $p = 0.819$ |
| 2 steps | 1.45 (1.08 to 1.95) $p = 0.015$ | 0.049 (0.004 to 0.095) $p = 0.035$ | 0.86 (0.74 to 0.99) $p = 0.039$ | −0.024 (−0.050 to 0.001) $p = 0.060$ |
| 3 steps | 1.18 (0.85 to 1.63) $p = 0.325$ | 0.019 (−0.024 to 0.063) $p = 0.386$ | 0.95 (0.71 to 1.27) $p = 0.716$ | −0.011 (−0.065 to 0.044) $p = 0.702$ |
| 4 steps | 0.92 (0.60 to 1.41) $p = 0.708$ | −0.013 (−0.070 to 0.044) $p = 0.657$ | 0.88 (0.70 to 1.09) $p = 0.240$ | −0.022 (−0.058 to 0.013) $p = 0.210$ |
| 5 steps | 0.97 (0.67 to 1.39) $p = 0.852$ | −0.005 (−0.047 to 0.036) $p = 0.798$ | 1.11 (0.91 to 1.36) $p = 0.294$ | 0.032 (−0.030 to 0.094) $p = 0.308$ |
| 6 steps | 0.41 (0.24 to 0.69) $p = 0.001$ | −0.069 (−0.131 to −0.006) $p = 0.031$ | 0.63 (0.53 to 0.76) $p < 0.001$ | −0.114 (−0.176 to −0.053) $p < 0.001$ |
| **Time step** | | | | |
| Step 0 | 1 (base) | 1 (base) | 1 (base) | 1 (base) |
| Step 1 | 1.41 (1.07 to 1.85) $p = 0.014$ | 0.031 (0.009 to 0.052) $p = 0.005$ | 0.76 (0.53 to 1.07) $p = 0.115$ | −0.030 (−0.068 to 0.008) $p = 0.121$ |
| Step 2 | 1.33 (0.75 to 2.35) $p = 0.328$ | 0.025 (−0.027 to 0.076) $p = 0.350$ | 1.09 (0.74 to 1.60) $p = 0.659$ | 0.010 (−0.040 to 0.060) $p = 0.684$ |
| Step 3 | 1.36 (0.93 to 1.99) $p = 0.109$ | 0.027 (−0.003 to 0.058) $p = 0.073$ | 1.25 (0.85 to 1.86) $p = 0.257$ | 0.031 (−0.024 to 0.086) $p = 0.266$ |
| Step 4 | 1.45 (0.91 to 2.30) $p = 0.114$ | 0.034 (−0.004 to 0.073) $p = 0.081$ | 1.35 (0.88 to 2.07) $p = 0.168$ | 0.043 (−0.016 to 0.102) $p = 0.150$ |
| Step 5 | 1.76 (1.17 to 2.66) $p = 0.007$ | 0.062 (0.022 to 0.101) $p = 0.002$ | 1.28 (1.11 to 1.48) $p = 0.001$ | 0.035 (0.016 to 0.054) $p < 0.001$ |
| Step 6 | 1.67 (1.09 to 2.56) $p = 0.019$ | 0.053 (0.011 to 0.094) $p = 0.012$ | 1.46 (1.05 to 2.03) $p = 0.025$ | 0.053 (0.004 to 0.102) $p = 0.033$ |
| Step 7 | 1.50 (0.81 to 2.75) $p = 0.195$ | 0.038 (−0.021 to 0.097) $p = 0.210$ | 2.66 (1.99 to 3.56) $p < 0.001$ | 0.197 (0.124 to 0.269) $p < 0.001$ |
| **Random effects (95% CI)** | | | | |
| **Clinic** | 0.0127 (0.0012 to 0.1348) | 0.0002 (0.0000 to 0.0030) | $1.74 \times 10^{-33}$ ($1.50 \times 10^{-33}$ to $2.02 \times 10^{-33}$) | $5.43 \times 10^{-34}$ ($9.78 \times 10^{-47}$ to $3.01 \times 10^{-21}$) |
| **Log pseudolikelihood** | −2,276.62 | −1,704.65 | −2,420.40 | −2,169.21 |

*(Continued)*

**Table 6.** (Continued)

| Variable | VL monitoring $n = 6,397$ | | Repeat HIV testing $n = 6,815$ | |
|---|---|---|---|---|
| | Model 1 | Model 1 | Model 1 | Model 1 |
| | RR (95% CI) *p*-Value | Absolute RD (95% CI) *p*-Value | RR (95% CI) *p*-Value | Absolute RD (95% CI) *p*-Value |
| Akaike information criteria | 4,565.24 | 3,421.30 | 4,852.79 | 4,350.43 |

Model 1 includes time-step fixed effects, clinic random effects, and cluster robust standard errors.

**Abbreviations:** CQI, continuous quality improvement; RD, risk difference; RR, relative risk; VL, HIV viral load

critical indicator of success in interrupting transmission of HIV from mother to child. In absolute terms, our effect size was relatively small on a per-visit basis, about 4 percentage points per visit. However, it is well recognised that multiple ANC visits are required for maternal and neonatal health [60]. For the median number of 6 ANC visits in this community, we estimate a cumulative absolute effect size of about 12 percentage points. This effect size is close to the threshold of 15 percentage points for clinically meaningful effects that we used in our power calculations.

We also found heterogeneous effects of CQI on VL monitoring by duration of the cluster's exposure to CQI: effects reached their maximum 2 to 4 months after intervention rollover and waned thereafter. The time pattern we see here can be explained by a "natural course" of innovation—initially health workers need to learn to implement innovations to maximum effectiveness; over time, the need for the innovations (e.g., novel processes supporting VL monitoring) declines as more and more patients have benefited. At the same time, the waning effects following a maximum could also indicate health worker fatigue in implementing an innovation that has now become routine. This latter explanation would argue for implementation research on approaches to maintain long-term innovation effects.

However, our findings also emphasise that it is important to better understand the contextual determinants and potential mechanisms of CQI effects, because CQI did not have an effect on our second primary endpoint, repeat HIV testing. There are several plausible reasons for the difference in CQI effects. The first reason is perceptions of clinical impact. For VL monitoring, impact is likely very salient—lower maternal mortality, better maternal health, and reduced MTCT. In contrast, for repeat HIV testing, impact may be perceived as more "distal" to clinical outcomes and thus less important. CQI activities aimed at increasing repeat HIV testing may therefore not have resonated as strongly with health workers as did CQI activities aimed at improving VL monitoring. A related issue is patient perceptions or behaviours. For example, women not living with HIV may have less perceived risk of infection [64] or concerns about negative partner attitudes if they received a positive HIV diagnosis [65]. Presenting late to ANC for the first time may also have been a barrier to receiving a second HIV test for some women, because the guidelines stipulate an interval of 3 months between HIV tests.

The second potential reason is the differential baseline attainment of our 2 primary endpoints—about one-third for VL monitoring and about two-thirds for repeat HIV testing. While our power calculations indicated that our trial was sufficiently powered to detect significant CQI effects on both primary endpoints, it might have been practically easier to identify and address problems in achieving high levels of coverage with VL monitoring compared to repeat HIV testing, because VL monitoring involves more processes and complexity than repeat HIV testing—thus, CQI that addresses process quality had fewer opportunities to change practice in repeat HIV testing. For instance, VL monitoring involved identifying

women eligible for the test, phlebotomy, following up and documenting results, and retrospectively capturing routine ART clinic data onto the national monitoring and evaluation database (TIER.Net)—CQI could address flaws in most of these processes. During each visit, women living with HIV may have interacted with different staff cadres in the clinic—nurses for clinical management or lay counsellors for adherence counselling—with additional opportunities to identify eligible women for testing during the clinic appointment. Other staff (e.g., data capturers) may have also identified women eligible for VL monitoring while filing previous VL results or capturing routine data onto TIER.Net after the visit. Conversely, repeat HIV testing involved a simpler workflow: identifying eligible women, conducting the rapid test, and immediately documenting the result, with only one type of staff cadre involved (usually a lay counsellor).

The final potential reason for the difference in CQI effects is that, during the study period, many of the HIV lay counsellors, who were carrying out HIV testing in primary care clinics, lost their employment in the public-sector health system in South Africa because of a lay counsellor re-deployment policy [66]. The human resources for HIV testing in primary care were thus diminished over the study period. In contrast, nurses carried out VL monitoring in the South African health system, and their numbers remained stable over the study period. The differential CQI effectiveness on our 2 primary endpoints may thus have been caused by underlying differences in the availability of the human resources needed to effectively implement endpoint-specific CQI activities.

The 2 other trials of CQI in primary care in sub-Saharan Africa did not find significant effects, likely due to selection of endpoints that were relatively distal to CQI activities. As alluded to previously, the variability of these endpoints—neonatal and perinatal mortality in Malawi [13] as well as a distal healthcare utilisation outcome, postpartum retention, in Nigeria [14]—may be a function of many factors that CQI cannot be expected to influence. Our endpoints were more proximate to CQI thereby making it easier to demonstrate a direct CQI effect.

In general, however, the 2 prior studies and ours jointly point towards the need to ensure sufficient resources and sufficiently good underlying health systems processes for CQI to be effective. For instance, staff turnover or redeployment during CQI initiatives are likely to impede CQI effectiveness, whereas clinic health worker willingness to engage with CQI, sufficient protected time for clinic health workers to work with CQI mentors, and relevance of CQI initiatives to local needs are likely enablers of CQI effectiveness [13, 14, 67].

## CQI implementation and resource investments

We observed that health workers maintained positive attitudes towards CQI and our CQI mentors during the study. In particular, health workers were enthusiastic to learn CQI tools and improve quality of services and were keen to continue working with the CRH team.

CQI is a systematic process for identifying problems and devising, testing, and revising potential solutions to quality shortcomings. The different management techniques used during this process are highly standardised and time-tested, including process mapping, fishbone diagrams, and PDSA cycles. Deviations from the overall process and these individual techniques —by the CQI mentors implementing them or by clinic health workers utilising them—would imply low fidelity. Based on our observations, CQI mentors implemented the CQI intervention with high fidelity. However, clinic health workers' implementation of the CQI process and techniques was of somewhat lower fidelity. Clinic health workers could not always participate in CQI meetings and failed to complete some of the CQI tasks, despite their general enthusiasm for the intervention. The reasons for these deviations were mostly rooted in

structural capacity constraints in this rural South African health system, including staffing shortages, rapid staff turnover, and competing clinical commitments [68].

The CQI process and management techniques are intended to be the same across clinics and communities; the solutions that they identify, test, and revise are intended to be tailored to each clinic and will therefore differ across contexts. The solutions that each clinic identified and implemented did indeed differ, even though all participating clinics were located in the same district in rural KwaZulu-Natal. We will report further details on these results in an upcoming publication on the process evaluation that accompanied this trial.

The key resource needed to implement CQI is CQI mentors' and clinic health workers' time. CQI mentors typically spent several hours per week training and supporting clinic health workers during the intensive phase of CQI; during the maintenance phase, the mentors spent substantially less time interacting with the clinic health workers, typically several hours per month. The training sessions at the very beginning of each step and the action learning sessions each required an entire day. CQI activities required time to (i) identify drivers of poor VL monitoring and repeat HIV testing, (ii) conduct group meetings to design clinic-specific solutions, (iii) implement those solutions (e.g., different methods to track women eligible for HIV care tests), and (iv) use clinic-based real-time data sources to test whether particular solutions improved testing rates.

CQI mentors who are adaptable and relate professionally and culturally to health workers at target health facilities may be more successful at engaging health workers in CQI. Mentors must be adept at generating descriptive data and, ideally, be familiar with implementing the desired change in practice. Obtaining basic CQI certification may take between around 18 hours (online) [69] to 3 days [70] of coursework.

In addition to the CQI mentors' time and the health workers' time, a so-called improvement advisor—a more highly qualified health professional than the CQI mentors with the ability to coach and lead the CQI mentors—is an integral part of the team to oversee CQI activities. In our experience, the improvement advisor would need to invest around 20% of their time for successful CQI support; in other settings, this would depend on the needs of the CQI mentors and participating clinics. Given these CQI resource needs, the time—and financial—costs of the intervention were overall low. Feasibility and scalability of CQI will, however, not only depend on resource requirements but also on clinic health workers' motivation and mindset. CQI requires changes in individual behaviours and institutional practices, which in turn are determined by job satisfaction, ambition, experience, and local workplace culture and constraints [40].

In our study, the CQI mentors typically held CQI meetings 2 to 3 times per week during the intensive 2-month step. For routine implementation at larger scale, alternative meeting frequencies may also work. For instance, it may be more practicable to conduct visits less frequently (e.g., once per week) but over a longer period of time (e.g., half a year). The optimal frequency of mentor visits will likely depend on how many clinic health workers are available for CQI activities and whether the clinic activities that the CQI team chose for improvement require additional support.

## Strengths and limitations

Our pragmatic cluster-randomised stepped-wedge design rigorously tested the effectiveness of CQI in a real-world setting in rural South Africa while enabling all clinics to eventually receive the intervention. Our registered primary endpoints were not only important process indicators of HIV care quality expected to respond to CQI, but they were also essential clinical practice standards described in the national South African guidelines for HIV treatment and ANC. The

intervention was implemented flexibly by adapting to local clinic needs, and individual participant inclusion criteria were broad. Our CQI implementation team of local CQI mentors, improvement advisor, data manager, and scientific partners were from the University of Kwa-Zulu-Natal, had extensive experience implementing CQI elsewhere in South Africa, and spoke the vernacular language, isiZulu. The research was co-led by South African researchers, while most researchers in the wider scientific team have worked in rural South Africa for many years.

Our study had several limitations. First, 2-month intervention steps may have been too short to catalyse the changes required to improve outcomes given that clinic resources in routine care are already overstretched. However, the monthly maintenance visits following the intervention steps may have compensated for the relatively short initial implementation period. Second, we measured our endpoints as documented in the maternity case records. Although most maternity case records contained clinical information, the poor documentation of VL results among women who had a VL test performed raises concerns about data quality overall. It is thus possible that actual repeat HIV testing and VL monitoring rates were higher than measured in our study albeit not documented. In this case, we would also expect that the effect sizes that we have measured are underestimates of the true effect sizes. Third, we were unable to measure long-term sustainability of CQI over several years, because our study duration was limited by our research budget.

## Other policy and future research implications

In addition to the general policy relevance of our findings for health policies aiming to improve the quality of primary care in sub-Saharan Africa, our findings are of specific relevance to the goal of eMTCT. Universal coverage of the activities captured by our 2 primary endpoints—VL monitoring and repeat HIV testing in pregnancy—will be critical to achieve this goal [24]. In particular, VL monitoring close to delivery is key to ensuring maternal virologic suppression in the peripartum period [20], although VL monitoring and repeat HIV testing are crucial throughout pregnancy and breastfeeding to enable timely interventions for mother and baby. According to our findings, CQI could be a good addition to the set of interventions driving eMTCT in HIV hyperendemic settings.

Improving clinical processes to achieve desired health systems and health outcomes requires changes in practice (behavioural changes), which depend on individual motivation, job satisfaction, and clinical knowledge and context [40]. It is also unclear as to how far clinic health workers "normalise" CQI into their practice once the CQI mentors have completed an engagement, particularly given the human resource shortages and high staff turnover in the local health system. Knowledge of time to assimilation of CQI could inform future study designs that account for a suitable time lag before outcomes can be expected to change.

An important question for future research is whether this CQI intervention "spilled over" to ANC activities outside HIV care and to primary care outside ANC. Such spill-over effects could be positive or negative. On the one hand, the clinic health workers trained in CQI may have applied their new skills to other clinical services, resulting in further quality improvements. On the other hand, health workers invested time and energy in this CQI and, as a result, may have neglected clinical services outside the scope of this intervention. In future research, we plan to measure the effects of CQI on a broader set of endpoints, which will capture the quality of a wide range of ANC functions.

Outcome measures that lie on plausible causal pathways from an exposure to an outcome can be used to empirically confirm mechanistic hypotheses. CQI is an intervention that can work its way to outcomes through multiple pathways, because it empowers local health

workers to identify and implement those health systems improvements and innovations that are most promising in the particular contexts in which they work. In this case—and other complex interventions—local health workers can identify likely mechanisms connecting an intervention to outcomes. In the case of HIV VL monitoring and repeat HIV testing, proximate endpoints that lie on the causal pathways from CQI could include availability of test kits, presence of systems for patient tracking and tracing, health worker motivation to test, and patient demand for testing.

While our quality of care measures are good choices to test the effectiveness of a CQI intervention, the ultimate purpose of healthcare is to reduce morbidity and mortality. Future mathematical modelling studies should estimate the impact of CQI on health outcomes that could be achieved given the effect sizes we measured in this study and local estimates of VL monitoring in different communities in sub-Saharan Africa.

Our study findings are likely transferable to other resource-poor communities in southern and sub-Saharan Africa, where nurses are the primary providers of ANC and HIV treatment. Given the resource shortages and increasing demand on HIV services with rollout of universal ART for all people living with HIV, patient and laboratory results tracking are likely to become more challenging with paper-based methods. Conversely, access to electronic laboratory results would increase healthcare provider availability for direct clinical care while maximising opportunities for detecting and managing maternal HIV viraemia. It is thus plausible that technological innovations in clinic data and management systems could contribute to CQI effectiveness in improving HIV care.

## Conclusions

We showed that CQI in primary care can be effective in rural South Africa. We found a significant CQI effect on one of our 2 primary endpoints, VL monitoring among pregnant women living with HIV. However, the fact that CQI failed to improve our second primary endpoint indicates that CQI success is sensitive to the particular health system processes it is intended to address. CQI should be considered as an intervention to improve quality of primary care in rural African communities. Its use should be closely monitored to ensure that we further improve our understanding of factors influencing its success.

Preliminary results of this trial were presented at the Conference on Retroviruses and Opportunistic Infections (CROI) in March 2018, Boston, MA, USA.

## Supporting information

**S1 CONSORT Checklist. CONSORT checklist for stepped-wedge trials.** CONSORT, Consolidated Standards of Reporting Trials.
(DOCX)

**S1 Text. Additional methods.**
(DOCX)

**S1 Table. Characteristics of participating primary care clinics.**
(DOCX)

**S2 Table. Summary of clinic staffing and recruitment to clinic CQI team.** CQI, continuous quality improvement
(DOCX)

**S1 Fig. HIV prevalence by maternal age group at delivery.**
(TIF)

## Acknowledgments

The authors wish to thank all study participants and the South African National Department of Health partners for their support and engagement with the study, and the UKZN Centre for Rural Health for implementing the intervention. We thank all colleagues at AHRI for their support with project operations, including Research Nurse Manager Mr Siphephelo Dlamini. We also thank the AHRI Research Data Management team for their support, in particular Mr Sabelo Ntuli for creating the map of the study area. We also extend our thanks to the DSMB for this trial (Professor Landon Myer, Professor Hoosen Coovadia, Professor Anna Coutsoudis, and Ms Kathy Baisley) for their excellent and ongoing support and advice. Finally, we extend our gratitude and thanks to the late Scientia Professor David A. Cooper AC, for his contributions to developing this manuscript and PhD supervision of Dr Yapa.

## Author Contributions

**Conceptualization:** H. Manisha Yapa, Terusha Chetty, Philippa Matthews, Kobus Herbst, Deenan Pillay, Sally Wyke, Till Bärnighausen.

**Data curation:** H. Manisha Yapa, Dickman Gareta, Kobus Herbst, Till Bärnighausen.

**Formal analysis:** H. Manisha Yapa, Jan-Walter De Neve, Awachana Jiamsakul, Pascal Geldsetzer, Guy Harling, Frank Tanser, Till Bärnighausen.

**Funding acquisition:** Terusha Chetty, Till Bärnighausen.

**Investigation:** H. Manisha Yapa, Carina Herbst, Wendy Dhlomo-Mphatswe, Mosa Moshabela, Till Bärnighausen.

**Methodology:** H. Manisha Yapa, Carina Herbst, Pascal Geldsetzer, Wendy Dhlomo-Mphatswe, Mosa Moshabela, Kobus Herbst, Sally Wyke, Till Bärnighausen.

**Project administration:** H. Manisha Yapa, Till Bärnighausen.

**Resources:** H. Manisha Yapa, Carina Herbst.

**Software:** H. Manisha Yapa, Carina Herbst, Dickman Gareta, Kobus Herbst.

**Supervision:** Frank A. Post, Awachana Jiamsakul, Deenan Pillay, Till Bärnighausen.

**Validation:** H. Manisha Yapa, Till Bärnighausen.

**Visualization:** H. Manisha Yapa, Jan-Walter De Neve.

**Writing – original draft:** H. Manisha Yapa.

**Writing – review & editing:** H. Manisha Yapa, Jan-Walter De Neve, Terusha Chetty, Carina Herbst, Frank A. Post, Awachana Jiamsakul, Pascal Geldsetzer, Guy Harling, Wendy Dhlomo-Mphatswe, Mosa Moshabela, Philippa Matthews, Osondu Ogbuoji, Frank Tanser, Dickman Gareta, Kobus Herbst, Deenan Pillay, Sally Wyke, Till Bärnighausen.

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
