## [Decision Letter · Decision Letter 0]

10 Nov 2019

Dear Dr. Yapa,

Thank you very much for submitting your manuscript "Can Continuous Quality Improvement increase coverage of antenatal HIV care tests in rural South Africa? Results of a stepped-wedge cluster-randomised controlled implementation trial" (PMEDICINE-D-19-03348) for consideration at PLOS Medicine. 

Your paper was discussed among the editorial team and sent to independent reviewers, including a statistical reviewer. The reviews are appended at the bottom of this email and any accompanying reviewer attachments can be seen via the link below:

[LINK]

In light of these reviews, we will not be able to accept the manuscript for publication in the journal in its current form, but we would like to invite you to submit a revised version that fully addresses the reviewers' and editors' comments. You will appreciate that we cannot make a decision about publication until we have seen the revised manuscript and your response, and we expect to seek re-review by one or more of the reviewers. 

We hope to receive your revised manuscript by Nov 29 2019 11:59PM. Please email us (plosmedicine@plos.org) if you have any questions or concerns.

Please let me know if you have any questions. Otherwise, we look forward to receiving your revised manuscript in due course. 

Sincerely,

Richard Turner, PhD

rturner@plos.org

In your data statement, rather than specifying "reasonable" requests please adapt the wording to "Requests from researchers who meet the criteria for access to confidential data ..." or similar. PLOS' data policy does not allow article authors to be the primary point of contact for data, so please modify the statement to note that contact should be made via the data repository. 

Please remove the rhetorical question from the title, and add a colon before the study descriptor.

Please add summary demographic information for stud participants to your abstract. 

The final sentence of the "methods and findings" subsection of your abstract should summarize the study's main limitations. 

At line 32, please begin the "conclusions" subsection of your abstract with "In this study, we found that ..." or similar. 

After the abstract, we will need to ask you to add a new and accessible "author summary" section in non-identical prose. You may find it helpful to consult one or two recent research papers published in PLOS Medicine to get a sense of the preferred style. 

Please remove the sentence of discussion at line 102, or move this to the discussion section. 

To your methods section, please add a brief mention of ethics approval and the situation regarding informed consent. 

Please avoid making claims of your study being "the first", and where a claim of primacy is made please add "to our knowledge" or similar. 

We ask you to adapt the first paragraph of the discussion section of your main text so that this provides a summary of the study's findings. 

Please remove trade marks from your ms. 

Please revise your reference list so that all entries match journal style. Italics and boldface text should be converted into plain text, for example; a maximum of six author names should be listed followed by "et al.". 

Please abbreviate journal names consistently; and add full access information to reference 8. 

Please provide an update on the status of reference 39. 

We ask you to convert the Tidier checklist into an individual supplementary file, referred to in the main text. 

Please also convert your CONSORT checklist into an individual supplementary file, referred to in the methods section of your main text. In the checklist, individual items should be referred to by section (e.g. "Methods") and paragraph number rather than by page or line numbers, as the latter generally change in the event of publication. 

Comments from the reviewers:

*** Reviewer #1: 

The process of evaluation of care (CQI) described here is extremely interesting and the authors / actors of this work should be congratulated for this approach. 

As a clinician, non-expert in this type of approach, I have only few "naive" questions:

1 / the quantitative results of VL at baseline were satisfying. Are the additional VL made thanks to the CQI at the same level?

2 / Why not choose the most important endpoint: "child infection" (or at least PCR positive at 6 weeks for example)?

Concerning the CQI itself:

3 / It is stated that the teams have welcomed the CQI procedure. Did the level of satisfaction remained similar after the CQI ?

4 / It would be interesting to specify in more detail the time and human ressources required for these interventions, in order to appreciate the feasibility in other contexts.

5 / Were the interventions homogeneous between the centers and during the study period ? Can we imagine a sort of "CQI of the CQI" ?

6 / After this procedure, can we consider a simplified procedure of CQI to be more routinely implemented ? (at least for this specific question of HIV in ANC)

7 / Did the CQI related improvement of care for HIV women, had an impact (favorable or deleterious or neutral) on other activities of ANC (outside HIV care )?

*** Reviewer #2: 

[See attachment]

Michael Dewey

*** Reviewer #3: 

This manuscript presents the results of a stepped-wedge cluster randomised trial in South Africa. The trial compared a CQI intervention to standard of care in nurse-led antenatal care facilities with the aim of improving viral load testing among women living with HIV, and repeat HIV testing among women who were HIV negative at presentation for antenatal care.

This is a clearly thought out, pragmatic study which is well described in the paper. The findings show that CQI can improve the completion of viral load testing during pregnancy in low-resource and high-burden settings. The lack of effect of repeat HIV testing is clearly considered and raises important issues around the broader context within which CQI interventions are implemented.

Overall, this is well written manuscript that offers important insights into both the potential for CQI interventions in primary care settings and the challenges with implementing and evaluating CQI interventions. I have a few minor comments and points of clarification.

P7 line 115: Study setting: here you say 6 clinics but the abstract and description said 7 clinics (6 clusters - 2 small clinics combined)

P11 data sources. If I understand correctly, both completion of maternal viral loads and repeat HIV testing were ascertained from the maternity case record retrospectively at the time of delivery. My experience is that the MCRs are very well completed but are there any concerns about data quality in these patient held records? What happened in the MCR was lost/had to be replaced and there was missing data for a period of the pregnancy? It might be useful to add something about the quality of the outcome data to the methods/discussion.

P14 line300/301: Unsure what is meant by HIV diagnosis coverage of HIV-positive women?

Line 306: Usually avoid starting sentence with number (50.3% …)

Line 341: missing a bracket

Discussion Line 364: I agree with the authors discussion of the providers perceived importance of repeat HIV testing compared to viral load. I wonder if there is also any differences from the patient side. Are there any issues with women refusing HIV testing or perhaps not being aware that it should happen so not asking if it isn't done? In contrast perhaps women living with HIV are more aware of the impact of having a viral load test.

Line 371-374: Not sure I clearly understand what the authors mean here. Do you mean the CQI intervention was practically easier, with the barriers to viral load testing easier to identify and address than the barriers to repeat HIV testing. Perhaps the CQI intervention "favoured" the viral load testing outcome in this way in so far as it was the lower hanging fruit of things to address in the clinics?

Lines 375-378: This is a really important issue. Staff turnover and buy-in pose a major threat to CQI interventions and this requires considering for the roll out of these interventions in low-resource health systems.

Lines 394-405: The authors discuss here the issue of selecting a more proximal outcome measure. Do the authors have any suggestions here for future research based on your experiences? What sorts of outcomes might be important to consider when evaluating and monitoring the success of CQI interventions?

Line 429/430: Is there a reason you do not account for a delay in intervention effect/time for the intervention to stabilise, perhaps in sensitivity analyses?

Lastly, the inclusion of the TIDieR framework is really helpful to think about how this or similar interventions could be scaled. The success of the intervention is dependent on the skills of the CQI mentors who deliver the intervention and how they engage with the health providers. Some discussion on issues around identifying/training suitable CQI mentors when scaling/reproducing this sort of intervention might be useful.

***

[LINK]

---

## [Decision Letter · Decision Letter 1]

6 May 2020

Dear Dr. Yapa,

Thank you very much for re-submitting your manuscript "The impact of Continuous Quality Improvement on coverage of antenatal HIV care tests in rural South Africa: results of a stepped-wedge cluster-randomised controlled implementation trial" (PMEDICINE-D-19-03348R1) for consideration at PLOS Medicine.

I have discussed the paper with editorial colleagues and it was also seen again by our reviewers. I am pleased to tell you that, provided the remaining editorial and production issues are fully dealt with, we expect to be able to accept the paper for publication in the journal.

[LINK]

We look forward to receiving the revised manuscript by May 13 2020 11:59PM. 

Sincerely,

Richard Turner, PhD

rturner@plos.org

Requests from Editors:

In your data statement, please briefly explain the reasons why data cannot be made available (e.g., restrictions imposed by the study ethics approval). 

Please quote the number of participants in each study arm in your abstract; and add a few words to note that no adverse events were reported. 

Please trim the author summary, ensuring that the three subsections contain 3-4 points, each of 1-2 short sentences.

Please remove the sentence of discussion at line 152.

In your reference list, please make sure that journal names are abbreviated consistently (including "PLoS Med.", for example). 

Please clarify the situation with regard to reference 65. 

Please rename the attachments with completed checklists (e.g. "S1_CONSORT_Checklist"). 

Comments from Reviewers:

*** Reviewer #1: 

the remarks made at first reading have all been taken into account and the answers are well argued. I have no additional comments.

*** Reviewer #2: 

The authors have addressed all my points.

Michael Dewey

*** Reviewer #3: 

The authors have thoroughly and thoughtfully responded to my comments and those raised by the other reviewers. I have no additional comments.

*** Reviewer #4: 

This reviewer did not review the original paper but two related things in my mind attenuate the potential impact of this manuscript. First, the CQI strategy is very sparsely described. The literature on QI across a large number of disease conditions (eg., stroke, HIV) as well as variety of settings and organizations is remarkably heterogenous in clinical effects. To advance the science of quality improvement strategies, research on such approaches must be crystal clear about the nature of the quality improvement intervention in order to make its effects (or absence therefore) interpretable. I did not see where in the manuscript enough detail on who the CRH team was composed of (credentials? training?), when were they supposed to goto the facilities (what was the dose?) and how much leeway they had; what the specific actions taken were (presumably training, mentoring, teaching etc). Second, the implementation outcomes are likewise underdeveloped particularly from a quantitative perspective. What fraction of intended mentoring or coaching visits were made? Who received the team? What evidence of implementation outcomes or process outcomes were present? Also, although not all of this is in the scope of perhaps this manuscript - implementation outcomes any all or any of these levels would not only be useful and interesting but indeed critical for making this study a contribution to the literature. Finally context is crutical 0 and although there are organizational scales that are difficult to use, even survey results like how leadership at the facility level perceived in QI activities could be useful. In addition to Proctors "specifying" implementation strategies guidance, this paper should also make use of standard for reporting quality improvement studies http://www.squire-statement.org/ . So in short, while I applaud the rigorous design, the impact on the field in general and on the CQI literature would be enhance with these considerations in my view. I would encourage addition of these data to results if possible, and if not, discussion and consideration for a future submission on these outcomes alone.

***

[LINK]

---

## [Editor Report · Decision Letter 2]

25 Jun 2020

Dear Dr Yapa, 

On behalf of my colleagues and the academic editor, Dr. Elvin H. Geng, I am delighted to inform you that your manuscript entitled "The impact of Continuous Quality Improvement on coverage of antenatal HIV care tests in rural South Africa: results of a stepped-wedge cluster-randomised controlled implementation trial" (PMEDICINE-D-19-03348R2) has been accepted for publication in PLOS Medicine. 

PRODUCTION PROCESS

PRESS

PROFILE INFORMATION

Thank you again for submitting the manuscript to PLOS Medicine. We look forward to publishing it. 

Best wishes, 

Richard Turner, PhD

Senior Editor 

PLOS Medicine

plosmedicine.org